# Multi-talker speech comprehension at different temporal scales in listeners with normal and impaired hearing

Jixing Li[1]*[†], Qixuan Wang[2,3][†], Qian Zhou[4], Lu Yang[4], Yutong Shen[5], Shujian Huang[5], Shaonan Wang[6], Liina Pylkkänen[7], Zhiwu Huang[4,8]*

[1]Department of Linguistics and Translation, City University of Hong Kong, Hong Kong, Hong Kong; [2]Department of Facial Plastic and Reconstructive Surgery, Eye and ENT Hospital, Fudan University, Shanghai, China; [3]ENT institute, Eye and ENT Hospital, Fudan University, Shanghai, China; [4]Department of Otolaryngology-Head and Neck Surgery, Shanghai Ninth People's Hospital, Shanghai Jiao Tong University School of Medicine, Shanghai, China; [5]Department of Computer Science and Technology, Nanjing University, Nanjing, China; [6]Institute of Automation, Chinese Academy of Sciences, Beijing, China; [7]Department of Linguistics, Department of Psychology, New York University, New York, United States; [8]College of Health Science and Technology, Shanghai Jiao Tong University School of Medicine, Shanghai, China

*For correspondence:
jixingli@cityu.edu.hk (JL);
huangzw86@126.com (ZH)

[†]These authors contributed equally to this work

Competing interest: The authors declare that no competing interests exist.

## eLife Assessment

This **valuable** study a computational language model, i.e., HM-LSTM, to quantify the neural encoding of hierarchical linguistic information in speech, and addresses how hearing impairment affects neural encoding of speech. Overall the evidence for the findings is **solid**, although the evidence for different speech processing stages could be strengthened by a more rigorous temporal response function (TRF) analysis. The study is of potential interest to audiologists and researchers who are interested in the neural encoding of speech.

**Abstract** Comprehending speech requires deciphering a range of linguistic representations, from phonemes to narratives. Prior research suggests that in single-talker scenarios, the neural encoding of linguistic units follows a hierarchy of increasing temporal receptive windows. Shorter temporal units like phonemes and syllables are encoded by lower-level sensory brain regions, whereas longer units such as sentences and paragraphs are processed by higher-level perceptual and cognitive areas. However, the brain's representation of these linguistic units under challenging listening conditions, such as a cocktail party situation, remains unclear. In this study, we recorded electroencephalogram (EEG) responses from both normal-hearing and hearing-impaired participants as they listened to individual and dual speakers narrating different parts of a story. The inclusion of hearing-impaired listeners allowed us to examine how hierarchically organized linguistic units in competing speech streams affect comprehension abilities. We leveraged a hierarchical language model to extract linguistic information at multiple levels—phoneme, syllable, word, phrase, and sentence—and aligned these model activations with the EEG data. Our findings showed distinct neural responses to dual-speaker speech between the two groups. Specifically, compared to normal-hearing listeners, hearing-impaired listeners exhibited poorer model fits at the acoustic, phoneme, and syllable levels, as well as the sentence levels, but not at the word and phrase levels. These results suggest that hearing-impaired listeners experience disruptions at both

shorter and longer temporal scales, while their processing at medium temporal scales remains unaffected.

## Introduction

Human speech encompasses elements at different levels, from phonemes to syllables, words, phrases, sentences, and paragraphs. These elements manifest over distinct timescales: Phonemes occur over tens of milliseconds, and paragraphs span a few minutes. Understanding how these units at different timescales are encoded in the brain during speech comprehension remains a challenge. In the visual system, it has been well established that there is a hierarchical organization such that neurons in the early visual areas have smaller receptive fields, while neurons in the higher-level visual areas receive inputs from lower-level neurons and have larger receptive fields (*Hubel and Wiesel, 1962*). This organizing principle is theorized to be mirrored in the auditory system, where a hierarchy of temporal receptive windows (TRW) extends from primary sensory regions to advanced perceptual and cognitive areas (*Hasson et al., 2008*; *Honey et al., 2012*; *Lerner et al., 2011*; *Murray et al., 2014*). Under this assumption, neurons in the lower-level sensory regions, such as the core auditory cortex, support rapid processing of the ever-changing auditory and phonemic information, whereas neurons in the higher cognitive regions, with their extended TRWs, process information at the sentence or discourse level.

Recent functional magnetic resonance imaging (fMRI) studies have shown some evidence that different levels of linguistic units are encoded at different cortical regions (*Blank and Fedorenko, 2020*; *Chang et al., 2022*; *Hasson et al., 2008*; *Lerner et al., 2011*; *Schmitt et al., 2021*). For example, *Schmitt et al., 2021*, used artificial neural networks to predict the next word in a story across five stacked timescales. By correlating model predictions with brain activity while listening to a story in an fMRI scanner, they discerned a hierarchical progression along the temporoparietal pathway. This pathway identifies the role of the bilateral primary auditory cortex in processing words over shorter durations and the involvement of the inferior parietal cortex in processing paragraph-length units over extended periods. Studies using electroencephalogram (EEG), magnetoencephalography (MEG), and electrocorticography (ECoG) have also revealed synchronous neural responses to different linguistic units at different timescales (e.g. *Ding et al., 2016*; *Ding and Simon, 2012*; *Honey et al., 2012*; *Luo and Poeppel, 2007*). Notably, *Ding et al., 2016*, showed that the MEG-derived cortical response spectrum concurrently tracked the time courses of abstract linguistic structures at the word, phrase, and sentence levels.

Although there is a growing consensus on the hierarchical encoding of linguistic units in the brain, the neural representations of these units in a multi-talker setting remain less explored. In the classic 'cocktail party' situation in which multiple speakers talk simultaneously (*Cherry, 1953*), listeners must separate a speech signal from a cacophony of other sounds (*McDermott, 2009*). Studies have shown that listeners with normal hearing can selectively attend to a chosen speaker in the presence of two competing speakers (*Brungart, 2001*; *Shinn-Cunningham, 2008*), resulting in enhanced neural responses to the attended speech stream (*Brodbeck et al., 2018*; *Ding and Simon, 2012*; *Mesgarani and Chang, 2012*; *O'Sullivan et al., 2015*; *Zion Golumbic et al., 2013*). For example, *Ding and Simon, 2012*, showed that neural responses were selectively phase-locked to the broadband envelope of the attended speech stream in the posterior auditory cortex. Furthermore, when the intensity of the attended and unattended speakers is separately varied, the neural representation of the attended speech stream adapts only to the intensity of the attended speaker. *Zion Golumbic et al., 2013*, further suggested that the neural representation appears to be more 'selective' in higher perceptual and cognitive brain regions such that there is no detectable tracking of ignored speech.

This selective entrainment to attended speech has primarily focused on the low-level acoustic properties of the speech, while largely ignoring the higher-level linguistic units beyond the phonemic levels. *Brodbeck et al., 2018*, were the first study to simultaneously compare the neural responses to the acoustic envelopes, as well as the phonemes and words in two competing speech streams. They found that although the acoustic envelopes of both the attended and unattended speech could be decoded from brain activity in the temporal cortex, only phonemes and words of the attended speech showed significant responses. However, as their study only examined two linguistic units, a complete

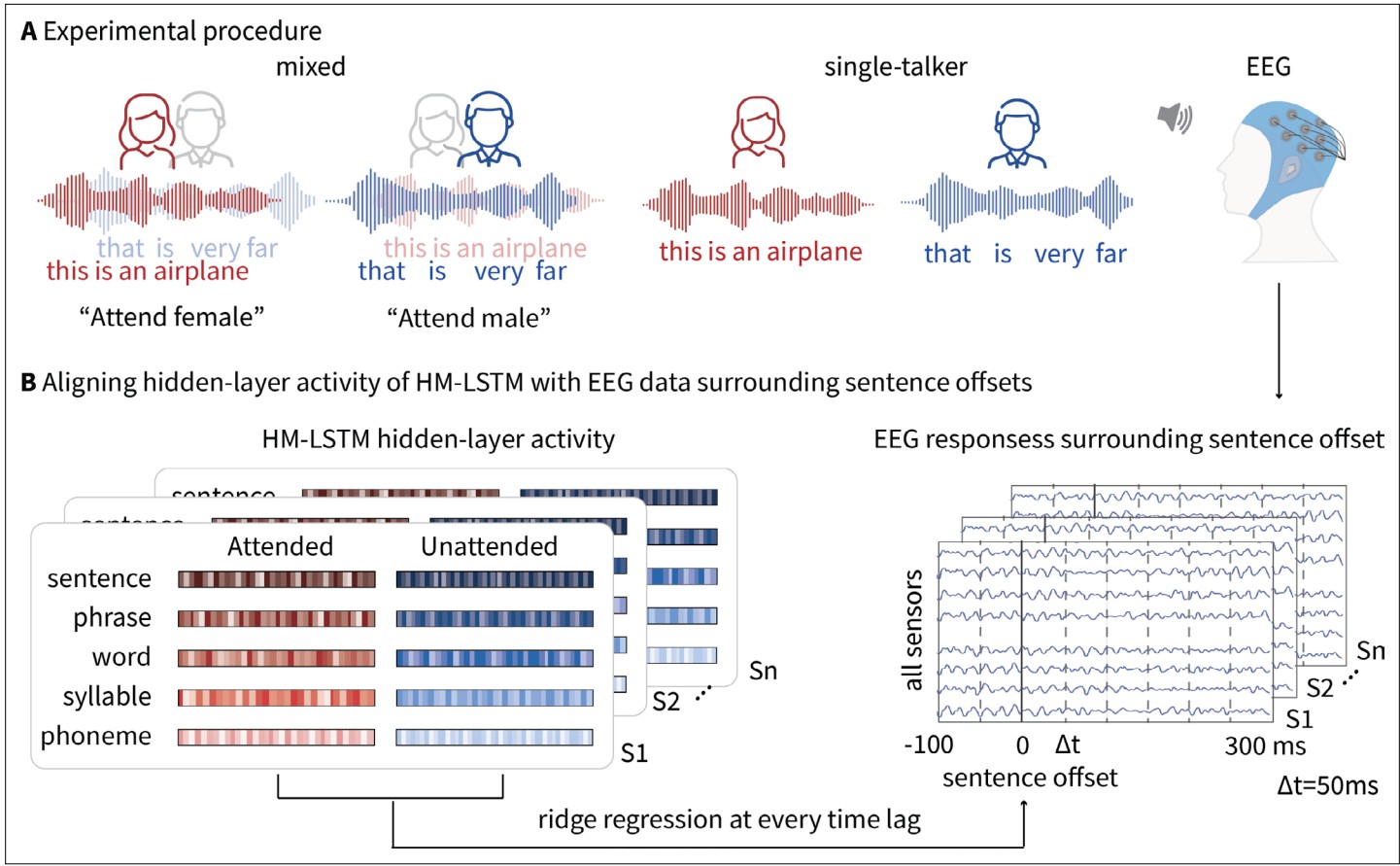

**Figure 1.** Methods and behavioral results. (**A**) Experimental procedure. The experimental task consisted of a multi-talker condition followed by a single-talker condition. In the multi-talker condition, the mixed speech was presented twice with the female and male speakers narrating simultaneously. Before each trial, instructions appeared in the center of the screen indicating which of the talkers to attend to (e.g. 'Attend female'). In the single-talker condition, the male and female speeches were presented sequentially. (**B**) Analyses pipeline. Hidden-layer activity of the hierarchical multiscale Long Short-Term Memory network (HM-LSTM) model, which represents each level of linguistic units for each sentence, was extracted and aligned with EEG data, time-locked to the offset of each sentence at nine different latencies.

The online version of this article includes the following figure supplement(s) for figure 1:

**Figure supplement 1.** Correlation matrices of regression outcomes for the five linguistic predictors between the electroencephalogram (EEG) data from delta, theta, and all frequency bands.

model of how the brain tracks linguistic units, from phonemes to sentences, in two competing speech streams is still lacking.

In this study, we investigate the neural underpinnings of processing diverse linguistic units in the context of competing speech streams among listeners with both normal and impaired hearing. We included hearing-impaired listeners to examine how hierarchically organized linguistic units in competing speech streams impact comprehension abilities. The experiment design consisted of a multi-talker condition and a single-talker condition. In the multi-talker condition, participants listened to mixed speech from female and male speakers narrating simultaneously. Before each trial, instructions on the screen indicated which speaker to focus on. In the single-talker condition, the speeches from male and female speakers were presented separately (refer to *Figure 1A* for the experimental procedure). We employed a hierarchical multiscale Long Short-Term Memory network (HM-LSTM; *Chung et al., 2017*) to dissect the linguistic information of the stimuli across phoneme, syllable, word, phrase, and sentence levels (detailed model architecture is described in the 'Hierarchical multiscale LSTM model' section in Materials and methods). We then performed ridge regressions using these linguistic units as regressors, time-locked to the offset of each sentence at nine latencies (see *Figure 1B* for the analysis pipeline). This model-brain alignment method has been commonly employed in the literature (e.g. *Caucheteux and King, 2022*; *Goldstein et al., 2022*; *Schmitt et al., 2021*; *Schrimpf*

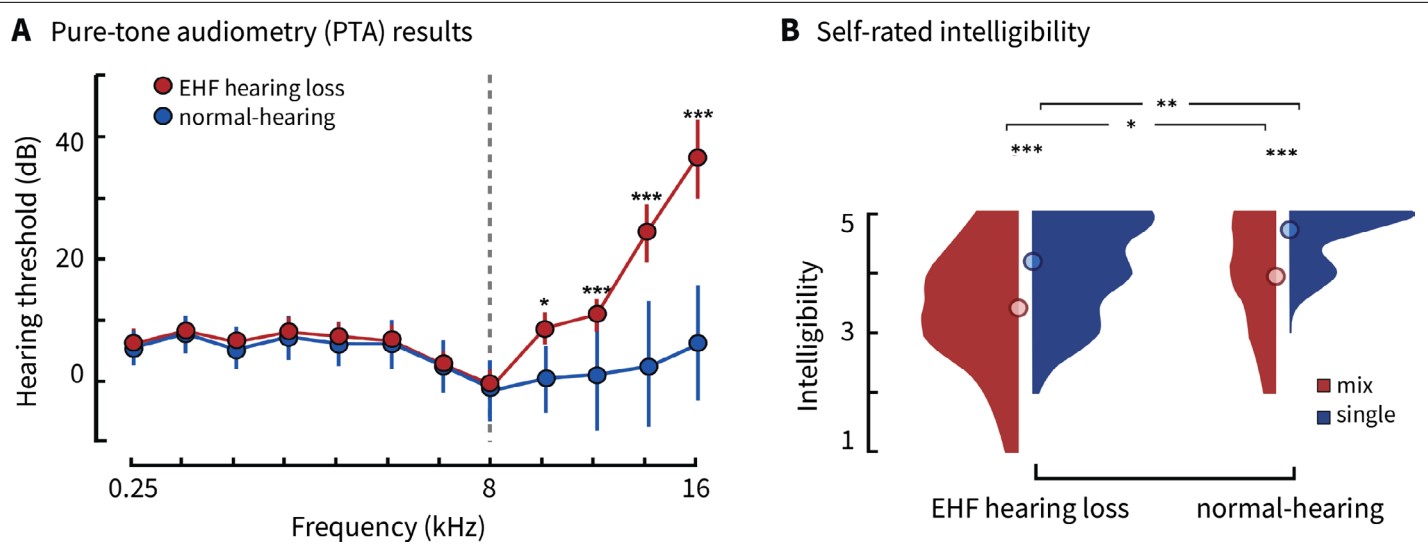

**Figure 2.** Behavioral results. (**A**) Pure tone audiometry (PTA) results for participants with normal hearing and extended high frequency (EHF) hearing loss. Starting at 10 kHz, participants with EHF hearing loss have significantly higher hearing thresholds (M=6.42 dB, SD = 7 dB) compared to normal-hearing participants (M=3.3 dB, SD = 4.9 dB; t=2, p=0.02). (**B**) Distribution of self-rated intelligibility scores for mixed- and single-talker speech across the two listener groups. * indicates p<0.05, ** indicates p<0.01, and *** indicates p<0.001.

*et al., 2021*). By examining how model alignments with brain activity vary across different linguistic levels between listeners with normal and impaired hearing, our goal is to identify specific levels of linguistic processing that pose comprehension challenges for hearing-impaired individuals.

## Results

### Behavioral results

A total of 41 participants (21 females, mean age = 24.1 years, SD = 2.1 years) with extended high frequency (EHF) hearing loss and 33 participants (13 females, mean age = 22.94 years, SD = 2.36 years) with normal hearing were included in the study. EHF hearing loss refers to hearing loss at frequencies above 8 kHz. It is considered a major cause of hidden hearing loss, which cannot be detected by audiometry (*Bharadwaj et al., 2019*). Although the phonetic information required for speech perception in quiet conditions is below 6 kHz, ample evidence suggests that salient information in the higher-frequency regions may also affect speech intelligibility (e.g. *Apoux and Bacon, 2004*; *Badri et al., 2011*; *Collins et al., 1981*; *Levy et al., 2015*). Unlike age-related hearing loss, EHF hearing loss is commonly found in young adults who frequently use earbuds and headphones for prolonged periods and are exposed to high noise levels during recreational activities (*Motlagh Zadeh et al., 2019*). Consequently, this demographic is a suitable comparison group for their age-matched peers with normal hearing. EHF hearing loss was diagnosed using the pure tone audiometry (PTA) test, thresholded at frequencies greater than 8 kHz. As shown in *Figure 2A*, starting at 10 kHz, participants with EHF hearing loss exhibit significantly higher hearing thresholds (M=6.42 dB, SD = 7 dB) compared to those with normal hearing (M=3.3 dB, SD = 4.9 dB), as confirmed by an independent two-sample one-tailed t-test (t(72)=2, p=0.02).

*Figure 2B* illustrates the distribution of intelligibility ratings for both mixed- and single-talker speech across the two listener groups. The average intelligibility ratings for both mixed- and single-talker speech were significantly higher for normal-hearing participants (mixed: M=3.89, SD = 0.83; single-talker: M=4.64, SD = 0.56) compared to hearing-impaired participants (mixed: M=3.38, SD = 1.04; single-talker: M=4.09, SD = 0.89), as shown by independent two-sample one-tailed t-tests (mixed: t(72)=2.11, p=0.02; single: t(72)=2.81, p=0.003). Additionally, paired two-sample one-tailed t-tests indicated significantly higher intelligibility scores for single-talker speech compared to mixed speech within both listener groups (normal-hearing: t(32)=4.58, p<0.0001; hearing-impaired: t(40)=4.28, p=0.0001). These behavioral results confirm that mixed speech presents greater comprehension

challenges for both groups, with hearing-impaired participants experiencing more difficulty in understanding both types of speech compared to those with normal hearing.

## HM-LSTM model performance

To simultaneously estimate linguistic content at the phoneme, syllable, word, phrase, and sentence levels, we adopted the HM-LSTM model originally developed by *Chung et al., 2017*. The original model consists of only two levels: the word level and the phrase level. We expanded its architecture to include five levels: phoneme, syllable, word, phrase, and sentence. Since our input consists of phoneme embeddings, we cannot directly apply their model, so we trained our model on the WenetSpeech corpus (*Zhang et al., 2021*), which provides phoneme-level transcripts. The inputs to the model were the vector representations of the phonemes in two sentences, and the output of the model was the classification result of whether the second sentence follows the first sentence (see *Figure 3A* and the 'Hierarchical multiscale LSTM model' section in 'Materials and methods' for the detailed model architecture). Unlike the transformer-based language models that can predict only word- or sentence- and paragraph-level information, our HM-LSTM model can disentangle the impact of different informational content associated with phonemic and syllabic levels. After 130 epochs, the model achieved an accuracy of 0.87 on the training data and 0.83 on our speech stimuli, which comprise 570 sentence pairs. We subsequently extracted activity from the trained model's four hidden layers for each sentence in our stimuli to represent information at the phoneme, syllable, word, sentence, and paragraph levels, with the sentence-level information represented by the last unit of the fourth layer. We computed the correlations among the activations at the five levels of the HM-LSTM model (see 'Correlations among LSTM model layers' section in 'Materials and methods' for the detailed analysis procedure). We did not observe very high correlations (all below 0.22) compared to prior model-brain alignment studies which report correlation coefficients above 0.5 for linguistic regressors (e.g. *Gao et al., 2024*; *Sugimoto et al., 2024*). In Chinese, a single syllable can also function as a word, potentially leading to higher correlations between regressors for syllables and words. However, we refrained from overinterpreting the results to suggest a higher correlation between syllable and sentence compared to syllable and word. A paired t-test of the syllable-word coefficients versus syllable-sentence coefficients across the 284 sentences revealed no significant difference (t(28399)=−3.96, p=1). This suggests that different layers of the model captured distinct patterns in the stimuli (see *Figure 3B*).

To verify that the hidden-layer activity indeed reflects information at the corresponding linguistic levels, we constructed a test dataset comprising 20 four-syllable sentences where all syllables contain the same vowels, such as 'mā ma mà m ǎ' (mother scolds horse) and 'shū shu sh ǔ shù' (uncle counts numbers). *Table 1* lists all four-syllable sentences with the same vowels. We hypothesized that the activity in the phoneme and syllable layer would be more similar than other layers for same-vowel sentences. The results confirmed our hypothesis: Hidden-layer activity for same-vowel sentences exhibited much more similar distributions at the phoneme and syllable levels compared to those at the word, phrase, and sentence levels. *Figure 3C* displays the scatter plot of the model activity at the five linguistic levels for each of the 20 four-syllable sentences, post dimension reduction using multidimensional scaling. We used color-coding to represent the activity of five hidden layers after dimensionality reduction. Each dot on the plot corresponds to one test sentence. Only phonemes are labeled because each syllable in our test sentences contains the same vowels (see *Table 1*). The plot reveals that model representations at the phoneme and syllable levels are more dispersed for each sentence, while representations at the higher linguistic levels—word, phrase, and sentence—are more centralized. Additionally, similar phonemes tend to cluster together across the phoneme and syllable layers, indicating that the model captures a greater amount of information at these levels when the phonemes within the sentences are similar.

## Regression results for single-talker speech versus attended speech

To examine the differences in neural activity between single- and dual-talker speech, we first compared the model fit of acoustic and linguistic features for the single-talker speech against the attended speech in the context of mixed speech across both listener groups (see 'Ridge regression at different time latencies' and 'Spatiotemporal clustering analysis' in 'Materials and methods' for analysis details). We have also analyzed the epoched EEG data by decomposing it into different frequency bands (see

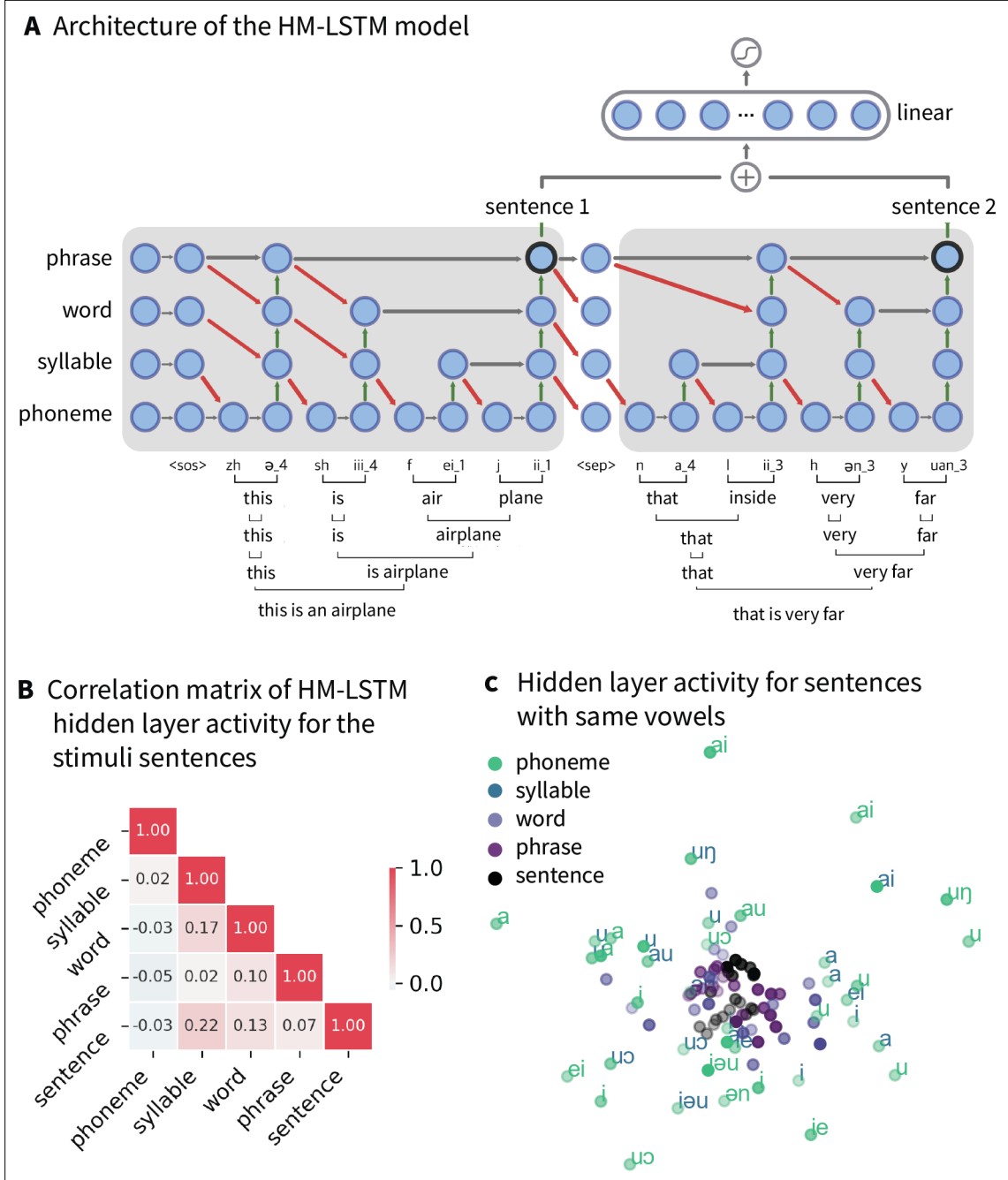

**Figure 3.** The hierarchical multiscale Long Short-Term Memory network (HM-LSTM) model architecture and hidden-layer activity for the stimuli sentences and the four-word Chinese sentences with same vowels. (**A**) The HM-LSTM model architecture. The model includes four hidden layers, corresponding to the phoneme-, syllable-, word-, and phrase-level information. Sentence-level information was represented by the last unit of the fourth layer. The inputs to the model were the vector representations of the phonemes in two sentences, and the output of the model was the classification result of whether the second sentence follows the first sentence. (**B**) Correlation matrix for the HM-LSTM model's hidden-layer activity for the sentences in the experimental stimuli. (**C**) Scatter plot of hidden-layer activity at the five linguistic levels for each of the 20 four-syllable sentences after multidimensional scaling (MDS).

'EEG recording and preprocessing' in 'Materials and methods'). We specifically examined the delta and theta bands, which are conventionally used in the literature for speech analysis. However, the results from these bands were very similar to those obtained using data from all frequency bands (see *Figure 1—figure supplement 1*). Therefore, we opted to use the epoched EEG data from all frequency bands for our analyses. *Figure 4* shows the sensors and time window where acoustic and linguistic

**Table 1.** All four-syllable Chinese sentences with same vowels.

| năi | nai | măi | mài |
|---|---|---|---|
| ʃə̆n | ʃən | ʃə̆n | ʃə̀n |
| gū̄ŋ | guŋ | tʃū̄ŋ | dùŋ |
| mèi | mei | méi | lèi |
| tɕiə̀u | tɕiəu | tɕʰiə̀u | tɕiə̀u |
| jí | ji | jí | jì |
| dì | ɕí | ɕīˉ | ɕì |
| tài | tai | b ă i | pāi |
| wá | wa | wā | wā |
| ʃū | fù | sù | dú |
| gū | fu | k ŭ | dú |
| bà | ba | dá | k ă |
| mā | ma | mà | m ă |
| buɔ́ | buɔ | buɔ́ | guɔˇ |
| ʃū | ʃu | ʃ ŭ | ʃù |
| gū | gu | b ŭ | bù |
| l ă ʊ | laʊ | náʊ | māʊ |
| puɔ́ | puɔ́ | duɔˇ | guɔ́ |
| tɕiě | tɕie | tɕié | tɕiē |
| dì | di | ɕ ĭ | dì |

features significantly better predicted EEG data in the left temporal region during single-talker speech as compared to the attended speech. It can be seen that for hearing-impaired participants, acoustic features showed a better model fit in single-talker settings as opposed to mixed speech conditions from –100 to 100 ms around sentence offsets (t=1.4, Cohen's d=1.5, p=0.002). However, no significant differences in the model fit between the single-talker and the attended speeches were observed for normal-hearing participants. Group comparisons revealed a significant difference in the model fit for the two conditions from –100 to 50 ms around sentence offsets (t=1.43, Cohen's d=1.28, p=0.011).

For the linguistic features, both the phoneme and syllable layers from the HM-LSTM model were more predictive of EEG data in single-talker speech compared to attended speech among hearing-impaired participants in the left temporal regions (phoneme: t=1.9, Cohen's d=0.49, p=0.004; syllable: t=1.9, Cohen's d=0.37, p=0.002). The significant effect occurred from approximately 0 to 100 ms for phonemes and 50 to 150 ms for syllables after sentence offsets. No significant differences in model fit were observed between the two conditions for participants with normal hearing. Comparisons between groups revealed significant differences in the contrast maps from 0 to 100 ms after sentence offsets for phonemes (t=2.39, Cohen's d=0.72, p=0.004) and from 50 to 150 ms after the sentence offsets for syllables (t=2.11, Cohen's d=0.78, p=0.001). The model fit to the EEG data for higher-level linguistic features—words, phrases, and sentences—does not show any significant differences between single-talker and attended speech across the two listener groups. This suggests that both normal-hearing and hearing-impaired participants are able to extract information at the word, phrase, and sentence levels from the attended speech in dual-speaker scenarios, similar to conditions involving only a single talker.

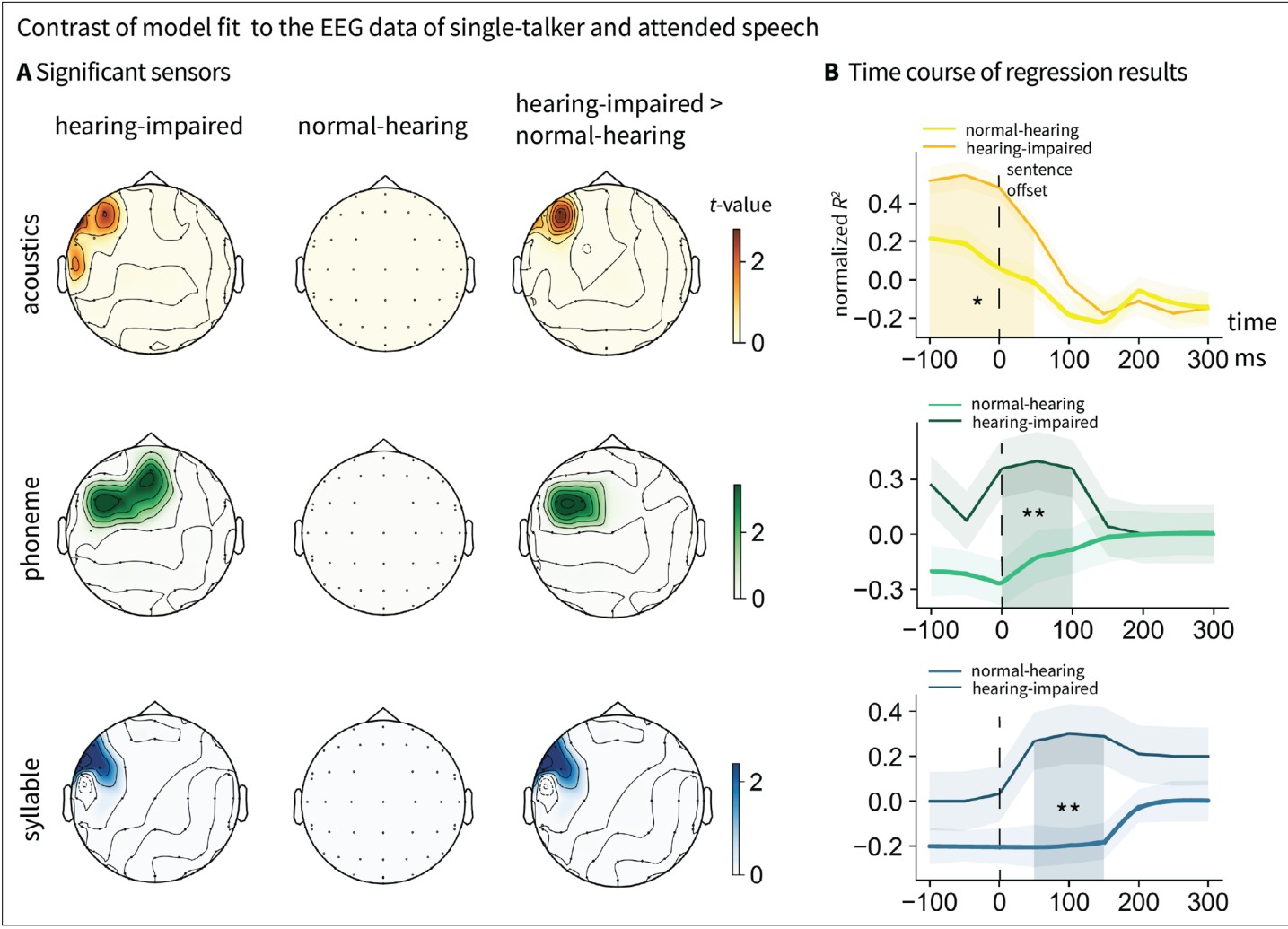

**Figure 4.** Significant sensor and time window for the model fit to the electroencephalogram (EEG) data for the acoustic and linguistic features extracted from the hierarchical multiscale Long Short-Term Memory network (HM-LSTM) model between single-talker and attended speech across the two listener groups. (**A**) Significant sensors showing higher model fit for single-talker speech compared to the attended speech at the acoustic, phoneme, and syllable levels for the two listener groups and their contrast. (**B**) Time courses of mean model fit in the significant clusters where normal-hearing participants showed higher model fit at the acoustic, phoneme, and syllable levels than hearing-impaired participants. The coefficient of determination ($R^2$) was z-transformed. Shaded regions indicate significant time windows. * denotes $p<0.05$, ** denotes $p<0.01$, and *** denotes $p<0.001$.

## Regression results for single-talker versus unattended speech

We also compared the model fit for single-talker speech and the unattended speech under the mixed speech condition. As shown in *Figure 5*, the acoustic features showed a better model fit in single-talker settings as opposed to mixed speech conditions from –100 to 50 ms around sentence offsets for hearing-impaired listeners (t=2.05, Cohen's d=1.1, p=<0.001) and from –100 to 50 ms for normal-hearing listeners (t=2.61, Cohen's d=0.23, p=<0.001). No group difference was observed with regard to the contrast of the model fit for the two conditions.

All the five linguistic features were more predictive of EEG data in single-talker speech compared to the unattended speech for both hearing-impaired participants (phoneme: t=1.72, Cohen's d=0.79, p<0.001; syllable: t=1.94, Cohen's d=0.9, p<0.001; word: t=2.91, Cohen's d=1.08, p<0.001; phrase: t=1.4, Cohen's d=0.61, p=0.041; sentence: t=1.67, Cohen's d=1.01, p=0.023) and normal-hearing participants (phoneme: t=1.99, Cohen's d=0.31, p=0.02; syllable: t=1.78, Cohen's d=0.8, p<0.001; word: t=2.85, Cohen's d=1.55, p=0.001; phrase: t=1.74, Cohen's d=1.4, p<0.001; sentence: t=1.86, Cohen's d=0.81, p=0.046). The significant effects occurred progressively later from phoneme to sentence level for both hearing-impaired participants (phoneme: –100–100 ms; syllable: 0–200

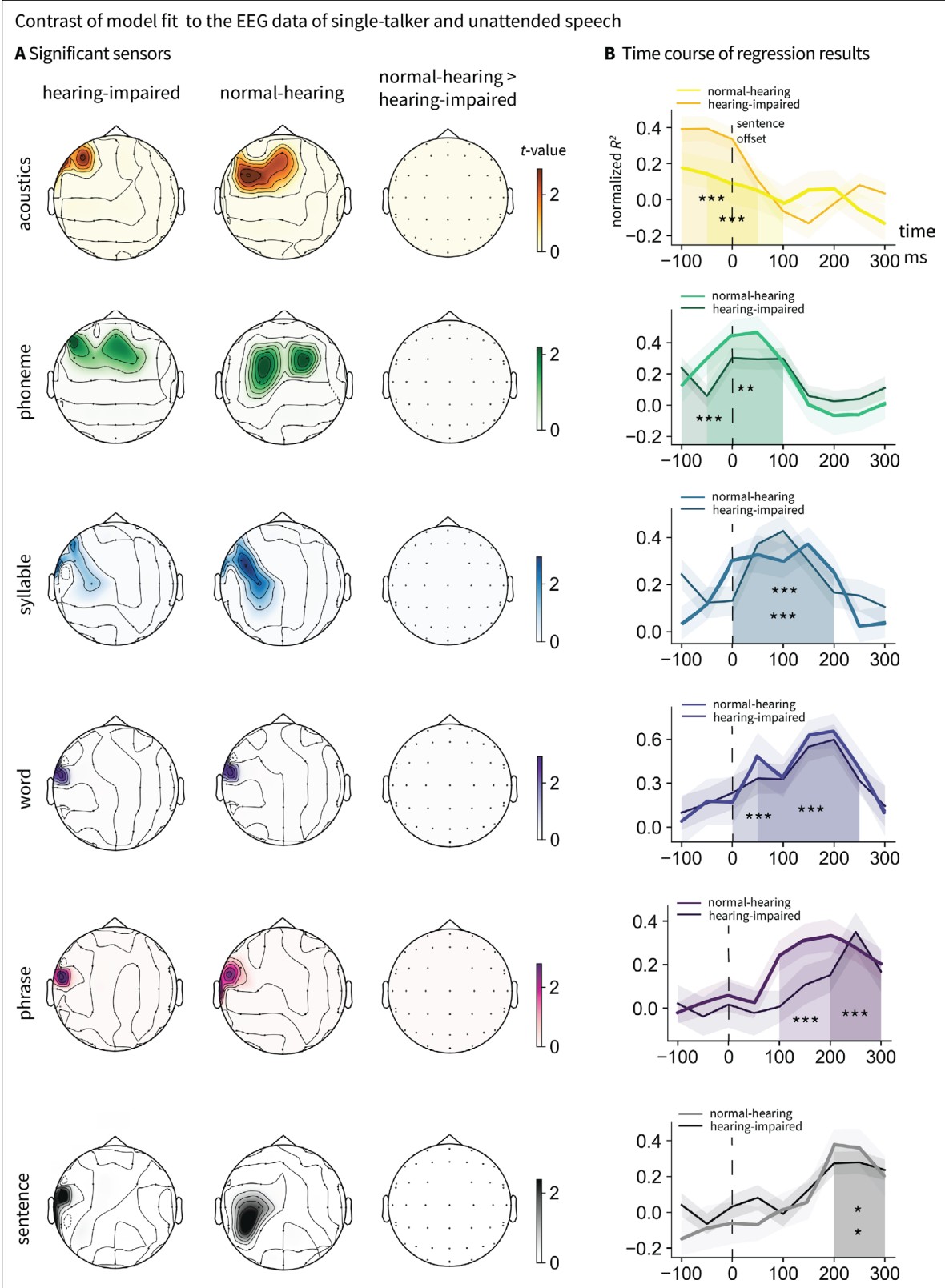

**Figure 5.** Significant sensor and time window for the model fit to the electroencephalogram (EEG) data for the acoustic and linguistic features between the single-talker and unattended speech in the mixed speech condition across the two listener groups. (**A**) Significant sensors showing higher model fit for the single-talker speech compared to the unattended speech at the acoustic and linguistic levels for the two listener groups and their contrast. (**B**) Time courses of mean model fit in the significant clusters. The significant time windows for within-group comparisons. The coefficient

*Figure 5 continued on next page*

*Figure 5 continued*

of determination ($R^2$) was z-transformed. Shaded regions indicate significant time windows. * denotes p<0.05, ** denotes p<0.01, and *** denotes p<0.001.

ms; word: 0–250 ms; phrase: 200–300 ms; sentence: 200–300 ms) and normal-hearing participants (phoneme: –50 to 100 ms; syllable: 0–200 ms; word: 50–250 ms; phrase: 100–300 ms; sentence: 200–300 ms). No significant group differences in the model fit were observed between the two conditions for all the linguistic levels.

## Regression results for attended versus unattended speech

*Figure 6* depicts the model fit of acoustic and linguistic predictors against EEG data for both attended and unattended speech while two speakers narrated simultaneously. It can be seen that for normal-hearing participants, acoustic features demonstrated a better model fit for attended speech compared to unattended speech from –100 ms to sentence offsets (t=3.21, Cohen's d=1.34, p=0.02). However, for hearing-impaired participants, no significant differences were observed in this measure. The difference between attended and unattended speech in normal-hearing and hearing-impaired participants was confirmed to be significant in the left temporal region from –100 to –50 ms before sentence offsets (t=2.24, Cohen's d=1.01, p=0.02) by a permutation two-sample t-test.

Both phoneme and syllable features significantly better predicted attended speech compared to unattended speech among normal-hearing participants (phoneme: t=1.58, Cohen's d=0.46, p=0.0006; syllable: t=1.05, Cohen's d=1.02, p=0.0001). The significant time window for phonemes was from –100 to 250 ms after sentence offsets, earlier than that for syllables, which was from 0 to 250 ms. No significant differences were observed in the hearing-impaired group. The contrast maps were significantly different across the two groups during the 0–100 ms window for phonemes (t=2.28, Cohen's d=1.32, p=0.026) and the 0–150 ms window for syllables (t=2.64, Cohen's d=1.04, p=0.022).

The word- and phrase-level features were significantly more effective at predicting EEG responses for attended speech than for unattended speech in both normal-hearing (word: t=2.59, Cohen's d=1.14, p=0.002; phrase: t=1.77, Cohen's d=0.68, p=0.027) and hearing-impaired listeners (word: t=3.61, Cohen's d=1.59, p=0.001; phrase: t=1.87, Cohen's d=0.71, p=0.004). The significant time windows for word processing were from 150 to 250 ms for hearing-impaired listeners and 150–200 ms for normal-hearing listeners. For phrase processing, significant time windows were from 150 to 300 ms for hearing-impaired listeners and 250–300 ms for normal-hearing listeners. No significant discrepancies were observed between the two groups regarding the model fit of words and phrases to the EEG data for attended versus unattended speeches. Surprisingly, we found a significantly better model fit for sentence-level features in attended speech for normal-hearing participants (t=1.52, Cohen's d=0.98, p=0.003) but not for hearing-impaired participants, and the contrast between the two groups was significant (t=1.7, Cohen's d=1.27, p<0.001), suggesting that hearing-impaired participants also struggle more with tracking information at longer temporal scales in multi-talker scenarios.

## mTRF results for attended versus unattended speech

To validate the linguistic features from our HM-LSTM models, we also examined baseline models for the linguistic features using multivariate temporal response function (mTRF) analysis. Our predictors include phoneme, syllable, word, phrase, and sentence (i.e. marking a value of 1 at each unit boundary offset). Our EEG data spans the entire 10 min duration for each condition, sampled at 10 ms intervals. The TRF results for our main comparison—attended versus unattended conditions—showed similar patterns to those observed using features from our HM-LSTM model (see *Figure 6—figure supplement 1*). At the phoneme and syllable levels, normal-hearing listeners showed marginally significantly higher TRF weights for attended speech compared to unattended speech at approximately –80 to 150 ms after phoneme offsets (t=2.75, Cohen's d=0.87, p=0.057) and 120 to 210 ms after syllable offsets (t=3.96, Cohen's d=0.73, p=0.083). At the word and phrase levels, normal-hearing listeners exhibited significantly higher TRF weights for attended speech compared to unattended speech at 190–290 ms after word offsets (t=4, Cohen's d=1.13, p=0.049) and around 120–290 ms after phrase offsets (t=5.27, Cohen's d=1.09, p=0.045). For hearing-impaired listeners, marginally significant effects were observed at 190–290 ms after word offsets (t=1.54, Cohen's d=0.6, p=0.059) and 180–290 ms after phrase offsets (t=3.63, Cohen's d=0.89, p=0.09). We also extracted the envelope at 10 ms intervals

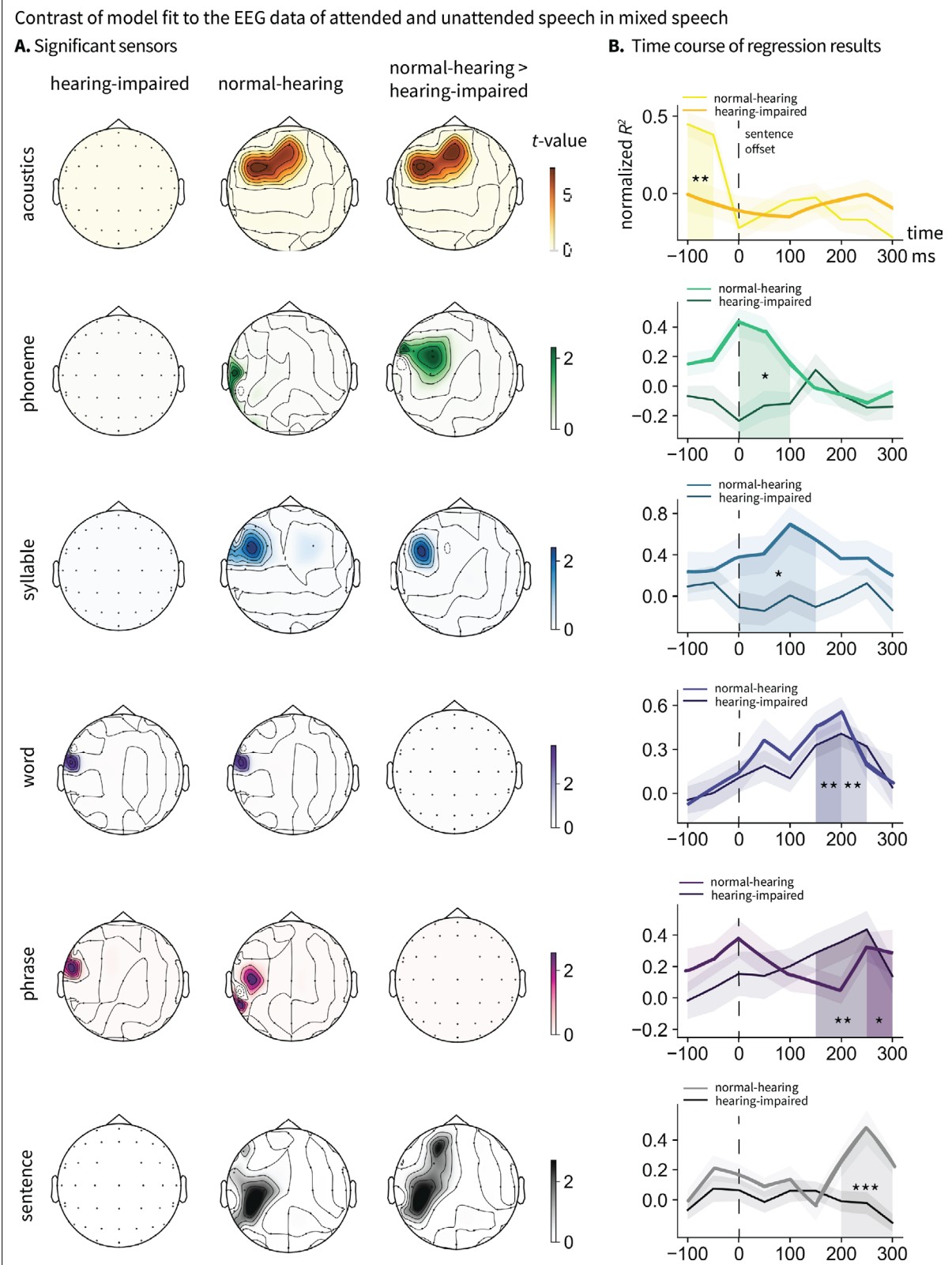

**Figure 6.** Significant sensor and time window for the model fit to the electroencephalogram (EEG) data for the acoustic and linguistic features between the attended and unattended speech in the mixed speech condition across the two listener groups. (**A**) Significant sensors showing higher model fit for the attended speech compared to the unattended speech at the acoustic and linguistic levels for the two listener groups and their contrast. (**B**) Time courses of mean model fit in the significant clusters where normal-hearing participants showed higher model fit than hearing-impaired participants. The

*Figure 6 continued on next page*

*Figure 6 continued*

coefficient of determination ($R^2$) was z-transformed. Shaded regions indicate significant time windows. * denotes p<0.05, ** denotes p<0.01, and *** denotes p<0.001.

The online version of this article includes the following figure supplement(s) for figure 6:

**Figure supplement 1.** Contrast of temporal response function (TRF) weights to the electroencephalogram (EEG) data of attended and unattended speech for the five linguistic predictors.

**Figure supplement 2.** Temporal response function (TRF) weights to the electroencephalogram (EEG) data of attended and unattended speech envelope.

for both attended and unattended speech and computed mTRFs independently for each subject and sensor using a basis of 50 ms Hamming windows spanning –100 to 300 ms relative to envelope onset. The results showed that in hearing-impaired participants, attended speech elicited a significant cluster in the bilateral temporal regions from 270 to 300 ms post-onset (t=2.40, p=0.01, Cohen's d=0.63). Unattended speech elicited an early cluster in right temporal and occipital regions from –100 to –80 ms (t=3.07, p=0.001, d=0.83). Normal-hearing participants showed significant envelope tracking in the left temporal region at 280–300 ms after envelope onset (t=2.37, p=0.037, d=0.48), with no significant cluster for unattended speech (see *Figure 6—figure supplement 2*).

## Discussion

Speech comprehension in a multi-talker environment is especially challenging for listeners with impaired hearing (*Fuglsang et al., 2020*). Consequently, exploring the neural underpinnings of multi-talker speech comprehension in hearing-impaired listeners could yield valuable insights into the challenges faced by both normal-hearing and hearing-impaired individuals in this scenario. Studies have reported abnormally enhanced responses to fluctuations in acoustic envelope in the central auditory system of older listeners (e.g. *Goossens et al., 2016*; *Parthasarathy et al., 2019*; *Presacco et al., 2016*) and listeners with peripheral hearing loss (*Goossens et al., 2018*; *Millman et al., 2017*). As older listeners also suffer from suppression of task-irrelevant sensory information due to reduced cortical inhibitory control functions (*Gazzaley et al., 2005*; *Gazzaley et al., 2008*), it is possible that impaired speech comprehension in a cocktail party situation arises from an attentional deficit linked to aging (*Du et al., 2016*; *Presacco et al., 2016*). However, younger listeners with EHF hearing loss have also reported difficulty understanding speech in a multi-talker environment (*Motlagh Zadeh et al., 2019*). It remains unknown what information is lost during multi-talker speech perception, and how the hierarchically organized linguistic units in competing speech streams affect the comprehension ability of people with impaired hearing.

In this study, we show that for normal-hearing listeners, the acoustic and linguistic features extracted from an HM-LSTM model can significantly predict EEG responses during both single-talker and attended speech in the context of two speakers talking simultaneously. Interestingly, their intelligibility scores for mixed speech are lower compared to single-talker speech, suggesting that normal-hearing listeners are still capable of tracking linguistic information at these levels in a cocktail party scenario, although with potentially greater effort. The model fit of the EEG data for attended speech is significantly higher than that for unattended speech across all levels. This aligns with previous research on 'selective auditory attention', which demonstrates that individuals can focus on specific auditory stimuli for processing, while effectively filtering out background noise (*Brodbeck et al., 2018*; *Brungart, 2001*; *Ding and Simon, 2012*; *Mesgarani and Chang, 2012*; *O'Sullivan et al., 2015*; *Shinn-Cunningham, 2008*; *Zion Golumbic et al., 2013*). Expanding on prior research which suggested that phonemes and words of attended speech could be decoded from the left temporal cortex of normal-hearing participants (*Brodbeck et al., 2018*), our results demonstrate that linguistic units across all hierarchical levels can be tracked in the neural signals.

For listeners with hearing impairments, the model fit for attended speech is significantly poorer at the acoustic, phoneme, and syllable levels compared to that for single-talker speech. Additionally, there is no significant difference in model fit at the acoustic, phoneme, and syllable levels between attended and unattended speech when two speakers are talking simultaneously. However, the model fit for the word and phrase features does not differ between single-talker and attended speech, and is significantly higher for that of the unattended speech. These findings suggest that hearing-impaired

listeners may encounter difficulties in processing information at shorter temporal scales, including the dynamic amplitude envelope and spectrotemporal details of speech, as well as phoneme- and syllable-level content. This is expected as our EHF hearing loss participants all exhibit higher hearing thresholds at frequencies above 8 kHz. Although these frequencies exceed those necessary for phonetic information in quiet environments, which are below 6 kHz, they may still impact the ability to process auditory information at faster temporal scales more than at slower speeds. Surprisingly, hearing-impaired listeners did not demonstrate an improved model fit for sentence features of the attended speech compared to the unattended speech, indicating that their ability to process information at longer temporal scales is also compromised. One limitation to consider is the absence of a behavioral task, such as comprehension questions at the end of the listening sections. This raises the possibility that the reduced cortical encoding of attended versus unattended speech across multiple linguistic levels in hearing-impaired listeners could stem from a different attentional strategy. For instance, they may focus on 'getting the gist' of the story or intermittently disengage from the task, tuning back in only for selected keywords or word combinations. However, we would like to emphasize that our hearing-impaired participants have EHF hearing loss, with impairment limited to frequencies above 8 kHz. This condition is unlikely to be severe enough to induce a fundamentally different attentional strategy for this task. Furthermore, normal-hearing listeners may also employ varied attentional strategies, yet the comparison still revealed significant differences. Based on these findings, we hypothesize that hearing-impaired listeners may struggle to extract low-level information from competing speech streams. Such a disruption in bottom-up processing could impede their ability to discern sentence boundaries effectively, which in turn hampers their ability to benefit from top-down information processing.

The hierarchical TRW hypothesis proposed that linguistic units at shorter temporal scales, such as phonemes, are encoded in the core auditory cortex, while information of longer duration is processed in higher perceptual and cognitive regions, such as the anterior temporal or posterior temporal and parietal regions (*Hasson et al., 2008*; *Honey et al., 2012*; *Lerner et al., 2011*; *Murray et al., 2014*). With the limited spatial resolution of EEG, we could not directly compare the spatial localization of these units at different temporal scales; however, we did observe an increasing latency in the significant model fit across different linguistic levels. Specifically, the significant time window for acoustic and phoneme features occurred around –100 to 100 ms relative to sentence offsets; syllables and words around 0–200 ms; and phrases and sentences around 200–300 ms. These progressively later effects from lower to higher linguistic levels suggest that these units may indeed be represented in brain regions with increasingly longer TRWs.

Our hierarchical linguistic contents were extracted using the HM-LSTM model adapted from *Chung et al., 2017*. This model has been adopted by *Schmitt et al., 2021*, to show that a 'surprisal hierarchy' based on the hidden-layer activity correlated with fMRI blood oxygen level-dependent signals along the temporal-parietal pathway during naturalistic listening. Although their research question is different from ours, their results suggested that the model has effectively captured information at different linguistic levels. Our testing results further confirmed that the model representations at the phoneme and syllable levels are different from model representations at the higher linguistic levels when the phonemes within the sentences are similar. Compared to the increasingly popular 'model-brain alignment' studies that typically use transformer architectures (e.g. *Caucheteux and King, 2022*; *Goldstein et al., 2022*; *Schrimpf et al., 2021*), our HM-LSTM model is considerably smaller in parameter size and does not match the capabilities of state-of-the-art large language models (LLMs) in downstream natural language processing (NLP) tasks such as question-answering, text summarization, translation, etc. However, our model incorporates phonemic and syllabic level representations, which are absent in LLMs that operate at the sub-word level. This feature could provide unique insights into how the entire hierarchy of linguistic units is processed in the brain.

It's important to note that we do not assert any similarity between the model's internal mechanisms and the brain's mechanisms for processing linguistic units at different levels. Instead, we use the model to disentangle linguistic contents associated with these levels. This approach has proven successful in elucidating language processing in the brain, despite the notable dissimilarities in model architectures compared to the neural architecture of the brain. For example, *Nelson et al., 2017*, correlated syntactic processing under different parsing strategies with the intracranial electrophysiological signals and found that the left-corner and bottom-up strategies fit the left temporal data better

than the most eager top-down strategy; *Goldstein et al., 2022*, and *Caucheteux and King, 2022*, also showed that the human brain and the deep learning language models share the computational principles as they process the same natural narrative.

In summary, our findings show that linguistic units extracted from a hierarchical language model better explain the EEG responses of normal-hearing listeners for attended speech, as opposed to unattended speech, when two speakers are talking simultaneously. However, hearing-impaired listeners exhibited poorer model fits at the acoustic, phoneme, syllable, and sentence levels, although their model fits at the word and phrase levels were not significantly affected. These results suggest that processing information at both shorter and longer temporal scales is especially challenging for hearing-impaired listeners when attending to a chosen speaker in a cocktail party situation. As such, these findings connect basic research on speech comprehension with clinical studies on hearing loss, especially hidden hearing loss, a global issue that is increasingly common among young adults.

## Materials and methods

### Participants
A total of 51 participants (26 females, mean age = 24 years, SD = 2.12 years) with EHF hearing loss and 51 normal-hearing participants (26 females, mean age = 22.92 years, SD = 2.14 years) took part in the experiment. 28 participants (18 females, mean age = 23.55, SD = 2.18) were removed from the analyses due to excessive motion, drowsiness, or inability to complete the experiment, resulting in a total of 41 participants (21 females, mean age = 24.1, SD = 2.1) with EHF hearing loss and 33 participants (13 females, mean age = 22.94, SD = 2.36) with normal hearing. All participants were right-handed native Mandarin speakers currently studying in Shanghai for their undergraduate or graduate degree, with no self-reported neurological disorders. EHF hearing loss was diagnosed using the PTA test, thresholded at frequencies above 8 kHz. The PTA was performed by experienced audiological technicians using an audiometer (Madsen Astera, GN Otometrics, Denmark) with headphones (HDA-300, Sennheiser, Germany) in a soundproof booth with background noise below 25 dB(A), as described previously (*Wang et al., 2021*). Air-conduction audiometric thresholds for both ears at frequencies of 0.5, 1, 2, 3, 4, 6, 8, 10, 12.5, 14, and 16 kHz were measured in 5 dB steps in accordance with the regulations of ISO 8253-1:2010.

### Stimuli
Our experimental stimuli were two excerpts from the Chinese translation of 'The Little Prince' (available at http://www.xiaowangzi.org/), previously used in fMRI studies where participants with normal hearing listened to the book in its entirety (*Li et al., 2022*). This material has been enriched with detailed linguistic predictions, from lexical to syntactic and discourse levels, using advanced NLP tools. Such rich annotation is critical for modeling hierarchical linguistic structures in our study. The two excerpts were narrated by one male and one female computer-synthesized voice, developed by the Institute of Automation, Chinese Academy of Sciences. The synthesized speech (available at https://osf.io/fjv5n/) is comparable to human narration, as confirmed by participants' post-experiment assessment of its naturalness. Additionally, using computer-synthesized voice instead of human-narrated speech alleviates the potential issue of imbalanced voice intensity and speaking rate that can arise between female and male narrators. The two sections were matched in length (approximately 10 min) and mean amplitude (approximately 65 dB), and were mixed digitally in a single channel to prevent any biases in hearing ability between the left and right ears.

### Experimental procedure
The experimental task consisted of a multi-talker condition and a single-talker condition. In the multi-talker condition, the mixed speech was presented twice with the female and male speakers narrating simultaneously. Before each trial, instructions appeared in the center of the screen indicating which of the talkers to attend to (e.g. 'Attend female'). In the single-talker condition, the male and female speeches were presented separately (see *Figure 1A* for the experiment procedure). The presentation order of the four conditions was randomized, and breaks were given between each trial. Stimuli were presented using insert earphones (ER-3C, Etymotic Research, USA) at a comfortable volume level of approximately 65 dB SPL. Participants were instructed to maintain visual fixation for the

duration of each trial on a crosshair centered on the computer screen and to minimize eye blinking and all other motor activities for the duration of each section. The whole experiment lasted for about 65 min, and participants rated the intelligibility of the multi-talker and the single-talker speeches on a five-point Likert scale after the experiment. The experiment was conducted at the Department of Otolaryngology-Head and Neck Surgery, Shanghai Ninth People's Hospital affiliated with the School of Medicine at Shanghai Jiao Tong University. The experimental procedures were approved by the Ethics Committee of the Ninth People's Hospital affiliated with Shanghai Jiao Tong University School of Medicine (SH9H-2019-T33-2). All participants provided written informed consent prior to the experiment and were paid for their participation.

## Acoustic features of the speech stimuli

The acoustic features included the broadband envelopes and the log-mel spectrograms of the two single-talker speech streams. The amplitude envelope of the speech signal was extracted using the Hilbert transform. The 129-dimension spectrogram and 1-dimension envelope were concatenated to form a 130-dimension acoustic feature at every 10 ms of the speech stimuli.

## Hierarchical multiscale LSTM model

We extended the original HM-LSTM model developed by *Chung et al., 2017*, to include not just the word and phrasal levels but also the sub-lexical phoneme and syllable levels. The model inputs were individual phonemes from two sentences, each transformed into a 1024-dimensional vector using a simple lookup table. This lookup table stores embeddings for a fixed dictionary of all unique phonemes in Chinese. This approach is a foundational technique in many advanced NLP models, enabling the representation of discrete input symbols in a continuous vector space. The output of the model was the classification result of whether the second sentence follows the first sentence. At each layer, the model implements a COPY or UPDATE operation at each time step t. The COPY operation maintains the current cell state without any changes until it receives a summarized input from the lower layer. The UPDATE operation occurs when a linguistic boundary is detected in the layer below, but no boundary was detected at the previous time step t–1. In this case, the cell updates its summary representation, similar to standard RNNs. We trained our model on 10,000 sentence pairs from the WenetSpeech corpus (*Zhang et al., 2021*), a collection that features over 10,000 hr of labeled Mandarin Chinese speech sourced from YouTube and podcasts. We used 1024 units for the input embedding and 2048 units for each HM-LSTM layer.

## Correlations among LSTM model layers

All the regressors are represented as 2048-dimensional vectors derived from the hidden layers of the trained HM-LSTM model. We applied the trained model to all 284 sentences in our stimulus text, generating a set of 284×2048-dimensional vectors. Next, we performed principal component analysis (PCA) on the 2048 dimensions and extracted the first 100 principal components (PCs), resulting in 284×100-dimensional vectors for each regressor. These 284×100 matrices were then flattened into 28,400-dimensional vectors. Subsequently, we computed the correlation matrix for the z-transformed 28,400-dimensional vectors of our five linguistic regressors.

## EEG recording and preprocessing

EEG was recorded using a standard 64-channel actiCAP mounted according to the international 10–20 system against a nose reference (Brain Vision Recorder, Brain Products). The ground electrode was set at the forehead. EEG signals were registered between 0.016 and 80 Hz with a sampling rate of 500 Hz. The impedances were kept below 20 kΩ. The EEG recordings were band-pass filtered between 0.1 and 45 Hz using a linear-phase finite impulse response filter. Independent component analysis was then applied to remove eye blink artifacts. The EEG data were then segmented into epochs spanning 500 ms pre-stimulus onset to 10 min post-stimulus onset and were subsequently downsampled to 100 Hz. We further decomposed the epoched EEG time series for each section into six classic frequency bands components (delta 1–3 Hz, theta 4–7 Hz, alpha 8–12 Hz, beta 12–20 Hz, gamma 30–45 Hz) by convolving the data with complex Morlet wavelets as implemented in MNE-Python (version 0.24.0). The number of cycles in the Morlet wavelets was set to frequency/4 for each frequency bin. The power values for each time point and frequency bin were obtained by taking the

square root of the resulting time-frequency coefficients. These power values were normalized to reflect relative changes (expressed in dB) with respect to the 500 ms pre-stimulus baseline. This yielded a power value for each time point and frequency bin for each section.

## Ridge regression at different time latencies

For each subject, we modeled the EEG responses at each sensor from the single-talker and mixed-talker conditions with our acoustic and linguistic features using ridge regression with a default regularization coefficient of 1 (see *Figure 1B*). For each sentence in the speech stimuli, we extracted the 2048-dimensional hidden-layer activity from

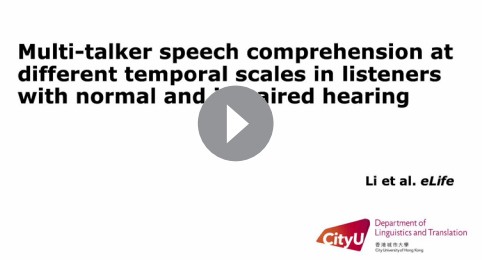

**Video 1.** Explanation of the ridge regression methods versus the multivariate temporal response function (mTRF) analyses. Explainer videos are not peer reviewewe used the python.
https://elifesciences.org/articles/100056/figures#video1

the HM-LSTM model to represent features at the phoneme, syllable, word, phrase, and sentence levels. We then employed PCA to reduce the 2048-dimensional vectors to the first 150 PCs. The 150-dimensional vectors for the five linguistic levels were then fit to the EEG signals time-locked to the offset of each sentence in the stimuli using ridge regression. The regressors are aligned to sentence offsets because our goal is to identify neural correlates for each model-derived feature of a whole sentence. If we align model activity with EEG data time-locked to sentence onsets, we would be finding neural responses to linguistic levels (from phoneme to sentence) of the whole sentence at the time when participants have not processed the sentence yet. Although this limits our analysis to a subset of the data (143 sentences×2 sections×400 ms windows), it targets the exact moment when full-sentence representations emerge against background speech, allowing us to examine each model-derived feature onto its neural signature. We understand that phonemes, syllables, words, phrases, and sentences differ in their durations. However, the five hidden activity vectors extracted from the model are designed to capture the representations of these five linguistic levels across *the entire sentence*. Specifically, for a sentence pair such as 'It can fly <sep > This is an airplane', the first 2048-dimensional vector represents all the phonemes in the two sentences ('t a_1 n əŋ_2 f ei_1<sep > zh ə_4 sh iii_4 f ei_1 j ii_1'), the second vector captures all the syllables ('ta_1 nəŋ_2 fei_1<sep > zhə_4 shiii_4 fei_1jii_1'), the third vector represents all the words, the fourth vector captures the phrases, and the fifth vector represents the sentence-level meaning. In our dataset, input pairs consist of adjacent sentences from the stimuli (e.g. Sentence 1 and Sentence 2, Sentence 2 and Sentence 3, and so on), and for each pair, the model generates five 2048-dimensional vectors, each corresponding to a specific linguistic level. To identify the neural correlates of these model-derived features—each intended to represent the full linguistic level across a complete sentence—we focused on the EEG signal surrounding the completion of the second sentence rather than on incremental processing. Accordingly, we extracted epochs from −100 ms to +300 ms relative to the offset of the second sentence and performed ridge regression analyses using the five model features (reduced to 150 dimensions via PCA) at every 50 ms across the epoch.

To assess the temporal progression of the regression outcomes, we conducted the analysis at nine sequential time points, ranging from 100 ms before to 300 ms after the sentence offset, with a 50 ms interval. We chose this time window as lexical or phrasal processing typically occurs 200 ms after stimulus offsets (*Bemis and Pylkkänen, 2011*; *Li et al., 2024*; *Li and Pylkkänen, 2021*). Additionally, we included the −100 to 200 ms time period in our analysis to examine phoneme- and syllable-level processing (e.g. *Gwilliams et al., 2022*). Using the entire sentence duration was not feasible, as the sentences in the stimuli vary in length, making statistical analysis challenging. Additionally, extending the window (e.g. to 2 s) would risk overlapping adjacent sentences. This would introduce ambiguity as to whether the EEG responses correspond to the current or the adjacent sentences. Additionally, our model activity represents a 'condensed final representation' at the five linguistic levels for the whole sentence, rather than incrementally during the sentence. We think the −100 to 300 ms time window relative to each sentence offset targets the exact moment when full-sentence representations are comprehended, and a 'condensed final representation' for the whole sentence across five

linguistic levels has been formed in the brain. The same regression procedure was applied to the 130-dimensional acoustic features (see *Video 1* for a more detailed description of the ridge regression methods).

Note that we did not use Pearson's r as in some prior studies using ridge regression as brain encoding models (e.g., *Goldstein et al., 2022*) because our analysis did not involve a train-test split. Specifically, *Goldstein et al., 2022*, divided their data into training and testing sets, trained a ridge regression model on the training set, and then used the trained model to predict neural responses on the test set. They calculated Pearson's r to assess the correlation between the predicted and observed neural responses, making the correlation coefficient (r) their primary measure of model performance. In contrast, our analysis focused on computing the model fitting performance ($R^2$) of the ridge regression model for each sensor and time point for each subject. At the group level, we conducted one-sample t-tests with spatiotemporal cluster-based correction on the $R^2$ values to identify sensors and time windows where $R^2$ values were significantly greater than baseline. We established the baseline by normalizing the $R^2$ values using Fisher z-transformation across sensors within each subject. The ridge regression was performed using customary Python codes, making heavy use of the scikit-learn (1.2.2) package.

## Spatiotemporal clustering analysis

The time courses of the z-transformed coefficient of determination ($R^2$) from the regression results at the nine time points for each sensor, corresponding to the linguistic and acoustic regressor for each subject, underwent spatiotemporal cluster permutation tests to determine their statistical significance at the group level. For instance, to assess whether words from the attended stimuli better predict EEG signals during the mixed speech compared to words from the unattended stimuli, we used the 150-dimensional vectors corresponding to the word layer from our LSTM model for the attended and unattended stimuli as regressors. We then fit these regressors to the EEG signals at nine time points (spanning –100 to 300 ms around the sentence offsets, with 50 ms intervals). We then conducted one-tailed two-sample t-tests to determine whether the differences in the contrasts of the z-transformed $R^2$ time courses were statistically significant. We repeated these procedures 10,000 times, replacing the observed t-values with shuffled t-values for each participant to generate a null distribution of t-values for each sensor. Sensors whose t-values were in the top 5th percentile of the null distribution were deemed significant (sensor-wise significance $p < 0.05$). The same method was applied to analyze the contrasts between attended and unattended speech during mixed speech conditions, both within and between groups. All our analyses were performed using custom Python codes, making heavy use of the mne (v.1.6.1), torch (v2.2.0), scipy (v1.12.0), and scikit-learn (1.2.2) packages.

## mTRF analysis

To validate the linguistic features from our HM-LSTM models, we also examined baseline models for the linguistic features using mTRF analysis. Our predictors include phoneme, syllable, word, phrase, and sentence (i.e. marking a value of 1 at each unit boundary offset). Our EEG data spans the entire 10 min duration for each condition, sampled at 10 ms intervals. Since our speech stimuli were computer-synthesized, the phoneme and syllable boundaries were automatically generated. The word boundaries were manually annotated by a native Mandarin speaker, as in *Li et al., 2022*. The phrase boundaries were automatically annotated by the Stanford parser (*Levy and Manning, 2003*) and manually checked by a native Mandarin speaker. These rate models are represented as five distinct binary time series, each aligned with the timing of the corresponding linguistic unit, making them well suited for mTRF analysis. Note that the rate regressors only encode the timing of linguistic unit boundaries, while the model-derived features encode the representational content of the linguistic input. We also extracted the envelope at 10 ms intervals for both attended and unattended speech. We computed mTRFs independently for each subject and sensor using a basis of 50 ms Hamming windows spanning –100 to 300 ms relative to all linguistic boundaries and envelope onset. The mTRF analysis was conducted using the Python package Eelbrain (v.0.39.8).

## Acknowledgements

This work was supported by the CityU Start-up Grant 7020086 and CityU Strategic Research Grant 7200747 (Li, J) the National Natural Science Foundation of China 82201273 (Wang Q), Shanghai

Science and Technology Commission Grant 22Y11902000 (Huang Z) and award G1001 from NYUAD Institute, New York University Abu Dhabi (Pylkkänen, L).

## Additional information

### Funding

| Funder | Grant reference number | Author |
| --- | --- | --- |
| City University of Hong Kong | 7020086 | Jixing Li |
| National Natural Science Foundation of China | 82201273 | Qixuan Wang |
| City University of Hong Kong | 7200747 | Jixing Li |
| Shanghai Science and Technology Commission Grant | 22Y11902000 | Zhiwu Huang |
| New York University Abu Dhabi | G1001 | Liina Pylkkänen |

The funders had no role in study design, data collection and interpretation, or the decision to submit the work for publication.

### Author contributions

Jixing Li, Conceptualization, Data curation, Supervision, Visualization, Methodology, Writing – original draft, Project administration, Writing – review and editing; Qixuan Wang, Conceptualization, Funding acquisition, Writing – review and editing; Qian Zhou, Data curation, Writing – review and editing; Lu Yang, Data curation; Yutong Shen, Shujian Huang, Shaonan Wang, Methodology; Liina Pylkkänen, Conceptualization, Supervision; Zhiwu Huang, Data curation, Funding acquisition

### Author ORCIDs

Jixing Li https://orcid.org/0000-0002-5210-6224
Qixuan Wang https://orcid.org/0000-0003-4109-6400

### Ethics

The experimental procedures were approved by the Ethics Committee of the Ninth People's Hospital affiliated with Shanghai Jiao Tong University School of Medicine (SH9H-2019-T33-2). All participants provided written informed consent prior to the experiment and were paid for their participation.

Reviewer #1 (Public review): https://doi.org/10.7554/eLife.100056.4.sa1
Reviewer #3 (Public review): https://doi.org/10.7554/eLife.100056.4.sa2
Author response https://doi.org/10.7554/eLife.100056.4.sa3

## Additional files

### Supplementary files

MDAR checklist

### Data availability

All code and preprocessed data are available at https://osf.io/fjv5n/. The raw data are not publicly shared because they contain potentially identifiable information and are subject to ethical restrictions. Interested researchers may request access to the original raw data by contacting the corresponding author at jixingli@cityu.edu.hk and submitting a project proposal. All requests will be reviewed by the Institutional Review Board at City University of Hong Kong. Use of the data for commercial research is not permitted.

The following dataset was generated:

| Author(s) | Year | Dataset title | Dataset URL | Database and Identifier |
|---|---|---|---|---|
| Li J | 2026 | Multi-Talker Speech Comprehension | https://osf.io/fjv5n/ | Open Science Framework, fjv5n |

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
