## [Editor Report · eLife Assessment]

This **valuable** study a computational language model, i.e., HM-LSTM, to quantify the neural encoding of hierarchical linguistic information in speech, and addresses how hearing impairment affects neural encoding of speech. Overall the evidence for the findings is **solid**, although the evidence for different speech processing stages could be strengthened by a more rigorous temporal response function (TRF) analysis. The study is of potential interest to audiologists and researchers who are interested in the neural encoding of speech.

---

## [Referee Report · Reviewer #1 (Public review)]

The authors relate a language model developed to predict whether a given sentence correctly followed another given sentence to EEG recordings in a novel way, showing receptive fields related to widely used TRFs. In these responses (or "regression results"), differences between representational levels are found, as well as differences between attended and unattended speech stimuli, and whether there is hearing loss. These differences are found per EEG channel.

In addition to these novel regression results, which are apparently captured from the EEG specifically around the sentence stimulus offsets, the authors also perform a more standard mTRF analysis using a software package (Eelbrain) and TRF regressors that will be more familiar to researchers adjacent to these topics, which was highly appreciated for its comparative value. Comparing these TRFs with the authors' original regression results, several similarities can be seen. Specifically, response contrasts for attended versus unattended speaker during mixed speech, for the phoneme, syllable, and sentence regressors, are greater for normal-hearing participants than hearing-impaired participants for both analyses, and the temporal and spatial extents of the significant differences are roughly comparable (left-front and 0 - 200 ms for phoneme and syllable, and left and 200 - 300 ms for sentence).

The inclusion of the mTRF analysis is helpful also because some aspects of the authors' original regression results, between the EEG data and the HM-LSTM linguistic model, are less than clear. The authors state specifically that their regression analysis is only calculated in the -100 - 300 ms window around stimulus/sentence offsets. They clarify that this means that most of the EEG data acquired while the participants are listening to the sentences is not analyzed, because their HM-LSTM model implementation represents all acoustic and linguistic features in a condensed way, around the end of the sentence. Thus the regression between data and model only occurs where the model predictions exist, which is the end of the sentences. This is in contrast to the mTRF analysis, which seems to have been done in a typical way, regressing over the entire stimulus time, because those regressors (phoneme onset, word onset, etc.) exist over the entire sentence time. If my reading of their description of the HM-LSTM regression is correct, it is surprising that the regression weights are similar between the HM-LSTM model and the mTRF model.

However, the code that the authors uploaded to OSF seems to clarify this issue. In the file ridge_lstm.py, the authors construct the main regressor matrices called X1 and X2 which are passed to sklearn to do the ridge regression. This ridge regression step is calculated on the continuous 10-minute bouts of EEG and stimuli, and it is calculated in a loop over lag times, from -100 ms to 300 ms lag. These regressor matrices are initialized as zeros, and are then filled in two steps: the HM_LSTM model unit weights are read from numpy files and written to the matrices at one timepoint per sentence (as the authors describe in the text), and the traditional phoneme, syllable, etc. annotations are ALSO read in (from csv files) and written to the matrices, putting 1s at every timepoint of those corresponding onsets/offsets. Thus the actual model regressor matrix for the authors' main EEG results includes BOTH the HM_LSTM model weights for each sentence AND the feature/annotation times, for whichever of the 5 features is being analyzed (phonemes, syllables, words, phrases, or sentences).

So for instance, for the syllable HM_LSTM regression results, the regressor matrix contains: (1) the HM_LSTM model weights corresponding to syllables (a static representation, placed once per sentence offset time), AND (2) the syllable onsets themselves, placed as a row of 1s at every syllable onset time. And as another example, for the word HM_LSTM regression results, the regressor matrix contains: (1) the HM_LSTM model weights corresponding to words (a static representation, placed once per sentence offset time), AND (2) the word onsets themselves, placed as a row of 1s at every word onset time.

If my reading of the code is correct, there are two main points of clarification for interpreting these methods:

First, the authors' window of analysis of the EEG is not "limited" to 400 ms as they say; rather the time dimension of both their ridge regression results and their traditional mTRF analysis is simply lags (400 ms-worth), and the responses/receptive fields are calculated over the entire 10-minute trials. This is the normal way of calculating receptive fields in a continuous paradigm. The authors seem to be focusing on the peri-sentence offset time points because that is where the HM_LSTM model weights are placed in the regressor matrix. Also because of this issue, it is not really correct when the authors say that some significant effect occurred at some latency "after sentence offset". The lag times of the regression results should have the traditional interpretation of lag/latency in receptive field analyses.

Second, as both the traditional linguistic feature annotations and the HM_LSTM model weights are part of the regression for the main ridge regression results here, it is not known what the contribution specifically of the HM_LSTM portion of the regression was. Because the more traditional mTRF analysis showed many similar results to the main ridge regression results here, it seems probable that the simple feature annotations themselves, rather than the HM_LSTM model weights, are responsible for the main EEG results. A further analysis separating these two sets of regressors would shed light on this question.

---

## [Referee Report · Reviewer #3 (Public review)]

Summary:

The authors aimed to investigate how the brain processes different linguistic units (from phonemes to sentences) in challenging listening conditions, such as multi-talker environments, and how this processing differs between individuals with normal hearing and those with hearing impairments. Using a hierarchical language model and EEG data, they sought to understand the neural underpinnings of speech comprehension at various temporal scales and identify specific challenges that hearing-impaired listeners face in noisy settings.

Strengths:

Overall, the combination of computational modeling, detailed EEG analysis, and comprehensive experimental design thoroughly investigates the neural mechanisms underlying speech comprehension in complex auditory environments.

The use of a hierarchical language model (HM-LSTM) offers a data-driven approach to dissect and analyze linguistic information at multiple temporal scales (phoneme, syllable, word, phrase, and sentence). This model allows for a comprehensive neural encoding examination of how different levels of linguistic processing are represented in the brain.

The study includes both single-talker and multi-talker conditions, as well as participants with normal hearing and those with hearing impairments. This design provides a robust framework for comparing neural processing across different listening scenarios and groups.

Weaknesses:

The study tests only a single deep neural network model for extracting linguistic features, which limits the robustness of the conclusions. A lower model fit does not necessarily indicate that a given type of information is absent from the neural signal-it may simply reflect that the model's representation was not optimal for capturing it. That said, this limitation is a common concern for data-driven, correlation-based approaches, and should be viewed as an inherent caveat rather than a critical flaw of the present work.

---

## [Author Response]

The following is the authors’ response to the previous reviews

**eLife Assessment**
This valuable study combines a computational language model, i.e., HM-LSTM, and temporal response function (TRF) modeling to quantify the neural encoding of hierarchical linguistic information in speech, and addresses how hearing impairment affects neural encoding of speech. The analysis has been significantly improved during the revision but remain somewhat incomplete - The TRF analysis should be more clearly described and controlled. The study is of potential interest to audiologists and researchers who are interested in the neural encoding of speech.

We thank the editors for the updated assessment. In the revised manuscript, we have added a more detailed description of the TRF analysis on p. of the revised manuscript. We have also updated Figure 1 to better visualize the analyses pipeline. Additionally, we have included a supplementary video to illustrate the architecture of the HM-LSTM model, the ridge regression methods using the model-derived features, and mTRF analysis using the acoustic envelop and the binary rate models.

**Public Reviews:**

**Reviewer #1 (Public review):**
About R squared in the plots:The authors have used a z-scored R squared in the main ridge regression plots. While this may be interpretable, it seems non-standard and overly complicated. The authors could use a simple Pearson r to be most direct and informative (and in line with similar work, including Goldstein et al. 2022 which they mentioned). This way the sign of the relationships is preserved.

We did not use Pearson’s r as in Goldstein et al. (2022) because our analysis did not involve a train-test split, which was a key aspect of their approach. Specifically, Goldstein et al. (2022) divided their data into training and testing sets, trained a ridge regression model on the training set, and then used the trained model to predict neural responses on the test set. They calculated Pearson’s r to assess the correlation between the predicted and observed neural responses, making the correlation coefficient (r) their primary measure of model performance. In contrast, our analysis focused on computing the model fitting performance (R²) of the ridge regression model for each sensor and time point for each subject. At the group level, we conducted one-sample t-tests with spatiotemporal cluster-based correction on the R² values to identify sensors and time windows where R² values were significantly greater than baseline. We established the baseline by normalizing the R² values using Fisher z-transformation across sensors within each subject. We have added this explanation on p.13 of the revised manuscript.

About the new TRF analysis:The new TRF analysis is a necessary addition and much appreciated. However, it is missing the results for the acoustic regressors, which should be there analogous to the HM-LSTM ridge analysis. The authors should also specify which software they have utilized to conduct the new TRF analysis. It also seems that the linguistic predictors/regressors have been newly constructed in a way more consistent with previous literature (instead of using the HM-LSTM features); these specifics should also be included in the manuscript (did it come from Montreal Forced Aligner, etc.?). Now that the original HM-LSTM can be compared to a more standard TRF analysis, it is apparent that the results are similar.

We used the Python package Eelbrain to conduct the multivariate temporal response function (mTRF) analyses. As we previously explained in our response to R3, we did not apply mTRF to the acoustic features due to the high dimensionality of the input. Specifically, our acoustic representation consists of a 130-dimensional vector sampled every 10 ms throughout the speech stimuli (comprising a 129-dimensional spectrogram and a 1dimensional amplitude envelope). This led to interpreting the 130-dimensional TRF estimation difficult to interpret. A similar constraint applied to the hidden-layer activations from our HMLSTM model for the five linguistic features. After dimensionality reduction via PCA, each still resulted in 150-dimensional vectors. To address this, we instead used binary predictors marking the offset of each linguistic unit (phoneme, syllable, word, phrase, sentence). Since our speech stimuli were computer-synthesized, the phoneme and syllable boundaries were automatically generated. The word boundaries were manually annotated by a native Mandarin as in Li et al. (2022). The phrase boundaries were automatically annotated by the Stanford parser and manually checked by a native Mandarin speaker. These rate models are represented as five distinct binary time series, each aligned with the timing of the corresponding linguistic unit, making them well-suited for mTRF analysis. Although the TRF results from the 1-dimensional rate predictors and the ridge regression results from the high-dimensional HM-LSTM-derived features are similar, they encode different things: The rate regressors only encode the timing of linguistic unit boundaries, while the model-derived features encode the representational content of the linguistic input. Therefore, we do not consider the mTRF analyses to be analogous to the ridge regression analyses. Rather, these results complement each other and both provide informative results into the neural tracking of linguistic structures at different levels for the attended and unattended speech.

Since the TRF result for the continuous acoustic features also concerns R2, we have added an mTRF analysis where we fitted the one-dimensional speech envelope to the EEG. We extracted the envelope at 10 ms intervals for both attended and unattended speech and computed mTRFs independently for each subject and sensor using a basis of 50 ms Hamming windows spanning –100 ms to 300 ms relative to envelope onset. The results showed that in hearing-impaired participants, attended speech elicited a significant cluster in the bilateral temporal regions from 270 to 300 ms post-onset (t = 2.40, p = 0.01, Cohen’s d = 0.63). Unattended speech elicited an early cluster in right temporal and occipital regions from –100 ms to –80 ms (t = 3.07, p = 0.001, d = 0.83). Normal-hearing participants showed significant envelope tracking in the left temporal region at 280–300 ms after envelope onset (t = 2.37, p = 0.037, d = 0.48), with no significant cluster for unattended speech. These results further suggest that hearing-impaired listeners may have difficulty suppressing unattended streams. We have added the new TRF results for envelope to Figure S3 and the “mTRF results for attended and unattended speech” on p.7 and the “mTRF analysis” in Material and Methods of the revised manuscript.

The authors' wording about this suggests that these new regressors have a nonzero sample at each linguistic event's offset, not onset. This should also be clarified. As the authors know, the onset would be more standard, and using the offset has implications for understanding the timing of the TRFs, as a phoneme has a different duration than a word, which has a different duration from a sentence, etc.

In our rate‐model mTRF analyses, we initially labelled linguistic boundaries as “offsets” because our ridge‐regression with HM-LSTM features was aligned to sentence offsets rather than onsets. However, since each offset coincides with the next unit’s onset—and our regressors simply mark these transition points as 1—the “offset” and “onset” models yield identical mTRFs. To avoid confusion, we have relabeled “offset” as “boundary” in Figure S2.

As discussed in our prior responses, this design was based on the structure of our input to the HM-LSTM model, where each input consists of a pair of sentences encoded in phonemes, such as “t a_1 n əŋ_2 f ei_1

We understand that phonemes, syllables, words, phrases, and sentences differ in their durations. However, the five hidden activity vectors extracted from the model are designed to capture the representations of these five linguistic levels across the entire sentence. Specifically, for a sentence pair such as “It can fly

About offsets:TRFs can still be interpretable using the offset timings though; however, the main original analysis seems to be utilizing the offset times in a different, more confusing way. The authors still seem to be saying that only the peri-offset time of the EEG was analyzed at all, meaning the vast majority of the EEG trial durations do not factor into the main HM-LSTM response results whatsoever. The way the authors describe this does not seem to be present in any other literature, including the papers that they cite. Therefore, much more clarification on this issue is needed. If the authors mean that the regressors are simply time-locked to the EEG by aligning their offsets (rather than their onsets, because they have varying onsets or some such experimental design complexity), then this would be fine. But it does not seem to be what the authors want to say. This may be a miscommunication about the methods, or the authors may have actually only analyzed a small portion of the data. Either way, this should be clarified to be able to be interpretable.

We hope that our response in RE4, along with the supplementary video, has helped clarify this issue. We acknowledge that prior studies have not used EEG data surrounding sentence offsets to examine neural responses at the phoneme or syllable levels. However, this is largely due to a lack of model that represent all linguistic levels across an entire sentence. There is abundant work comparing model predictors with neural data time-locked to offsets because they mark the point at which participants has already processed the relevant information (Brennan, 2016; Brennan et al., 2016; Gwilliams et al., 2024, 2025). Similarly, in our model– brain alignment study, our goal is to identify neural correlates for each model-derived feature. If we correlate model activity with EEG data aligned to sentence onsets, we would be examining linguistic representations at all levels (from phoneme to sentence) of the whole sentence at the time when participants have not heard the sentence yet. Although this limits our analysis to a subset of the data (143 sentences × 400 ms windows × 4 conditions), it targets the exact moment when full-sentence representations emerge against background speech, allowing us to examine each model-derived feature onto its neural signature. We have added this clarification on p.12 of the revised manuscript.

**Reviewer #2 (Public review):**
This study presents a valuable finding on the neural encoding of speech in listeners with normal hearing and hearing impairment, uncovering marked differences in how attention to different levels of speech information is allocated, especially when having to selectively attend to one speaker while ignoring an irrelevant speaker. The results overall support the claims of the authors, although a more explicit behavioural task to demonstrate successful attention allocation would have strengthened the study. Importantly, the use of more "temporally continuous" analysis frameworks could have provided a better methodology to assess the entire time course of neural activity during speech listening. Despite these limitations, this interesting work will be useful to the hearing impairment and speech processing research community. The study compares speech-in-quiet vs. multi-talker scenarios, allowing to assess within-participant the impact that the addition of a competing talker has on the neural tracking of speech. Moreover, the inclusion of a population with hearing loss is useful to disentangle the effects of attention orienting and hearing ability. The diagnosis of high-frequency hearing loss was done as part of the experimental procedure by professional audiologists, leading to a high control of the main contrast of interest for the experiment. Sample size was big, allowing to draw meaningful comparisons between the two populations.

We thank you very much for your appreciation of our research and we have now added a more description of the mTRF analyses on p.13-14 of the revised manuscript.

An HM-LSTM model was employed to jointly extract speech features spanning from the stimulus acoustics to word-level and phrase-level information, represented by embeddings extracted at successive layers of the model. The model was specifically expanded to include lower level acoustic and phonetic information, reaching a good representation of all intermediate levels of speech. Despite conveniently extracting all features jointly, the HMLSTM model processes linguistic input sentence-by-sentence, and therefore only allows to assess the corresponding EEG data at sentence offset. If I understood correctly, while the sentence information extracted with the HM-LSTM reflects the entire sentence - in terms of its acoustic, phonetic and more abstract linguistic features - it only gives a condensed final representation of the sentence. As such, feature extraction with the HM-LSTM is not compatible with a continuous temporal mapping on the EEG signal, and this is the main reason behind the authors' decision to fit a regression at nine separate time points surrounding sentence offsets.

Yes, you are correct. As explained in RE4, the model generates five hidden-layer activity vectors, each intended to represent all the phonemes, syllables, words, phrases within the entire sentence (“a condensed final representation”). This is the primary reason we extract EEG data surrounding the sentence offsets—this time point reflects when the full sentence has been processed by the human brain. We assume that even at this stage, residual neural responses corresponding to each linguistic level are still present and can be meaningfully analyzed.

While valid and previously used in the literature, this methodology, in the particular context of this experiment, might be obscuring important attentional effects impacted by hearing-loss. By fitting a regression only around sentence-final speech representations, the method might be overlooking the more "online" speech processing dynamics, and only assessing the permanence of information at different speech levels at sentence offset. In other words, the acoustic attentional bias between Attended and Unattended speech might exist even in hearing-impaired participants but, due to a lower encoding or permanence of acoustic information in this population, it might only emerge when using methodologies with a higher temporal resolution, such as Temporal Response Functions (TRFs). If a univariate TRF fit simply on the continuous speech envelope did not show any attentional bias (different trial lengths should not be a problem for fitting TRFs), I would be entirely convinced of the result. For now, I am unsure on how to interpret this finding.

We agree and we have added the mTRF results using the rate models for the 5 linguistic levels in the prior revision. The rate model aligns with the boundaries of each linguistic unit at each level. As explained in RE3, the rate regressors encode the timing of linguistic unit boundaries, while the model-derived features encode the representational content of the linguistic input. The mTRF results showed similar patterns to those observed using features from our HM-LSTM model with ridge regression (see Figure S2). These results complement each other and both provide informative results into the neural tracking of linguistic structures at different levels for the attended and unattended speech.

We have also added TRF results fitting the envelope of attended and unattended speech at every 10 ms to the whole 10-minute EEG data at every 10 ms. Our results showed that in hearing-impaired participants, attended speech elicited a significant cluster in the bilateral temporal regions from 270 to 300 ms post-onset (t = 2.40, p = 0.01, Cohen’s d = 0.63). Unattended speech elicited an early cluster in right temporal and occipital regions from –100 ms to –80 ms (t = 3.07, p = 0.001, d = 0.83). Normal-hearing participants showed significant envelope tracking in the left temporal region at 280–300 ms after envelope onset (t = 2.37, p = 0.037, d = 0.48), with no significant cluster for unattended speech. These results further suggest that hearing-impaired listeners may have difficulty suppressing unattended streams. We have added the new TRF results for envelope to Figure S3 and the “mTRF results for attended and unattended speech” on p.7 and the “mTRF analysis” in Material and Methods of the revised manuscript.

Despite my doubts on the appropriateness of condensed speech representations and singlepoint regression for acoustic features in particular, the current methodology allows the authors to explore their research questions, and the results support their conclusions. This work presents an interesting finding on the limits of attentional bias in a cocktail-party scenario, suggesting that fundamentally different neural attentional filters are employed by listeners with highfrequency hearing loss, even in terms of the tracking of speech acoustics. Moreover, the rich dataset collected by the authors is a great contribution to open science and will offer opportunities for re-analysis.

We sincerely thank you again for your encouraging comments regarding the impact of our study.

**Reviewer #3 (Public review):**
Summary:The authors aimed to investigate how the brain processes different linguistic units (from phonemes to sentences) in challenging listening conditions, such as multi-talker environments, and how this processing differs between individuals with normal hearing and those with hearing impairments. Using a hierarchical language model and EEG data, they sought to understand the neural underpinnings of speech comprehension at various temporal scales and identify specific challenges that hearing-impaired listeners face in noisy settings.Strengths:Overall, the combination of computational modeling, detailed EEG analysis, and comprehensive experimental design thoroughly investigates the neural mechanisms underlying speech comprehension in complex auditory environments. The use of a hierarchical language model (HM-LSTM) offers a data-driven approach to dissect and analyze linguistic information at multiple temporal scales (phoneme, syllable, word, phrase, and sentence). This model allows for a comprehensive neural encoding examination of how different levels of linguistic processing are represented in the brain. The study includes both single-talker and multi-talker conditions, as well as participants with normal hearing and those with hearing impairments. This design provides a robust framework for comparing neural processing across different listening scenarios and groups.Weaknesses:The analyses heavily rely on one specific computational model, which limits the robustness of the findings. The use of a single DNN-based hierarchical model to represent linguistic information, while innovative, may not capture the full range of neural coding present in different populations. A low-accuracy regression model-fit does not necessarily indicate the absence of neural coding for a specific type of information. The DNN model represents information in a manner constrained by its architecture and training objectives, which might fit one population better than another without proving the non-existence of such information in the other group. It is also not entirely clear if the DNN model used in this study effectively serves the authors' goal of capturing different linguistic information at various layers. More quantitative metrics on acoustic/linguistic-related downstream tasks, such as speaker identification and phoneme/syllable/word recognition based on these intermediate layers, can better characterize the capacity of the DNN model.

We agree that, before aligning model representations with neural data, it is essential to confirm that the model encodes linguistic information at multiple hierarchical levels. This is the purpose of our validation analysis: We evaluated the model’s representations across five layers using a test set of 20 four-syllable sentences in which every syllable shares the same vowel—e.g., “mā ma mà mǎ” (mother scolds horse), “shū shu shǔ shù” (uncle counts numbers; see Table S1). We hypothesized that the activity in the phoneme and syllable layer would be more similar than other layers for same-vowel sentences. The results confirmed our hypothesis: Hidden-layer activity for same-vowel sentences exhibited much more similar distributions at the phoneme and syllable levels compared to those at the word, phrase and sentence levels Figure 3C displays the scatter plot of the model activity at the five linguistic levels for each of the 20 4-syllable sentences, post dimension reduction using multidimensional scaling (MDS). We used color-coding to represent the activity of five hidden layers after dimensionality reduction. Each dot on the plot corresponds to one test sentence. Only phonemes are labeled because each syllable in our test sentences contains the same vowels (see Table S1).The plot reveals that model representations at the phoneme and syllable levels are more dispersed for each sentence, while representations at the higher linguistic levels—word, phrase, and sentence—are more centralized. Additionally, similar phonemes tend to cluster together across the phoneme and syllable layers, indicating that the model captures a greater amount of information at these levels when the phonemes within the sentences are similar.

Apart from the DNN model, we also included the rate models which simply mark 1 at each unit boundaries across the 5 levels. We performed mTRF analyses with these rate models and found similar patterns to our ridge‐regression results with the DNN: (see Figure S2). This provides further evidence that the model reliably captures information across all five hierarchical levels.

Since EEG measures underlying neural activity in near real-time, it is expected that lower-level acoustic information, which is relatively transient, such as phonemes and syllables, would be distributed throughout the time course of the entire sentence. It is not evident if this limited time window effectively captures the neural responses to the entire sentence, especially for lower-level linguistic features. A more comprehensive analysis covering the entire time course of the sentence, or at least a longer temporal window, would provide a clearer understanding of how different linguistic units are processed over time.

We agree that lower-level linguistic features may be distributed throughout the whole sentence, however, using the entire sentence duration was not feasible, as the sentences in the stimuli vary in length, making statistical analysis challenging. Additionally, since the stimuli consist of continuous speech, extending the time window would risk including linguistic units from subsequent sentences. This would introduce ambiguity as to whether the EEG responses correspond to the current or the following sentence. Additionally, our model activity represents a “condensed final representation” at the five linguistic levels for the whole sentence, rather than incrementally during the sentence. We think the -100 to 300 ms time window relative to each sentence offset targets the exact moment when full-sentence representations are comprehended and a “condensed final representation” for the whole sentence across five linguistic level have been formed in the brain. We have added this clarification on p.13 of the revised manuscript.

**Recommendations for the authors:**

**Reviewer #1 (Recommendations for the authors):**
Here are some specifics and clarifications of my public review:Initially I was interpreting the R squared as a continuous measure of predicted EEG relative to actual EEG, based on an encoding model, but this does not appear to be correct. Thank you for pointing out that the y axis is z-scored R squared in your main ridge regression plots. However, I am not sure why/how you chose to represent this that way. It seems to me that a simple Pearson r would be most informative here (and in line with similar work, including Goldstein et al. 2022 that you mentioned). That way you preserve the sign of the relationships between the regressors and the EEG. With R squared, we have a different interpretation, which is maybe also ok, but I also don't see the point of z-scoring R squared. Another possibility is that when you say "z-transformed" you are referring to the Fisher transformation; is that the case? In the plots you say "normalized", so that sounds like a z-score, but this needs to be clarified; as I say, a simple Pearson r would probably be best.

We did not use Pearson’s r, as in Goldstein et al. (2022), because our analysis did not involve a train-test split, which was central to their approach. In their study, the data were divided into training and testing sets, and a ridge regression model was trained on the training set. They then used the trained model to predict neural responses on the held-out test set, and calculated Pearson’s r to assess the correlation between the predicted and observed neural responses. As a result, their final metric of model performance was the correlation coefficient (r). In contrast, our analysis is more aligned with standard temporal response function (TRF) approaches. We did not perform a train-test split; instead, we computed the model fitting performance (R²) of the ridge regression model at each sensor and time point for each subject. At the group level, we conducted one-sample t-tests with spatiotemporal cluster-based correction on the R² values to determine which sensors and time windows showed significantly greater R² values than baseline. To establish a baseline, we z-scored the R² values across sensors and time points, effectively centering the distribution around zero. This normalization allowed us to interpret deviations from the mean R² as meaningful increases in model performance and provided a suitable baseline for the statistical tests. We have added this clarification on p.13 of the revised manuscript.

Thank you for doing the TRF analysis, but where are the acoustic TRFs, analogous to the acoustic results for your HM-LSTM ridge analyses? And what tools did you use to do the TRF analysis? If it is something like the mTRF MATLAB toolbox, then it is also using ridge regression, as you have already done in your original analysis, correct? If so, then it is pretty much the same as your original analysis, just with more dense timepoints, correct? This is what I meant by referring to TRFs originally, because what you have basically done originally was to make a 9-point TRF (and then the plots and analyses are contrasts of pairs of those), with lags between -100 and 300 ms relative to the temporal alignment between the regressors and the EEG, I think (more on this below).Also with the new TRF analysis, you say that the regressors/predictors had "a value of 1 at each unit boundary offset". So this means you re-made these predictors to be discrete as I and reviewer 3 were mentioning before (rather than using the HM-LSTM model layer(s)), and also, that you put each phoneme/word/etc. marker at its offset, rather than its onset? I'm also confused as to why you would do this rather than the onset, but I suppose it doesn't change the interpretation very much, just that the TRFs are slid over by a small amount.

We used the Python package Eelbrain (https://eelbrain.readthedocs.io/en/r0.39/auto_examples/temporal-response-functions/trf_intro.html) to conduct the multivariate temporal response function (mTRF) analyses. As we previously explained in our response to Reviewer 3, we did not apply mTRF to the acoustic features due to the high dimensionality of the input. Specifically, our acoustic representation consists of a 130-dimensional vector sampled every 10 ms throughout the speech stimuli (comprising a 129-dimensional spectrogram and a 1-dimensional amplitude envelope). This renders the 130 TRF weights to the acoustic features uninterpretable. However, we have now added TRF results from the 1- dimension envelope to the attended and unattended speech at every 10 ms.

A similar constraint applied to the hidden-layer activations from our HM-LSTM model for the five linguistic features. After dimensionality reduction via PCA, each still resulted in 150-dimensional vectors, further preventing their use in mTRF analyses. To address this, we instead used binary predictors marking the offset of each linguistic unit (phoneme, syllable, word, phrase, sentence). These rate models are represented as five distinct binary time series, each aligned with the timing of the corresponding linguistic unit, making them well-suited for mTRF analysis. It is important to note that these rate predictors differ from the HM-LSTMderived features: They encode only the timing of linguistic unit boundaries, not the content or representational structure of the linguistic input. Therefore, we do not consider the mTRF analyses to be equivalent to the ridge regression analyses based on HM-LSTM features

For onset vs. offset, as explained RE4, we labelled them “offsets” because our ridge‐regression with HM-LSTM features was aligned to sentence offsets rather than onsets (see RE4 and RE15 below for the rationale of using sentence offset). However, since each unit offset coincides with the next unit’s onset—and the rate model simply mark these transition points as 1—the “offset” and “onset” models yield identical mTRFs. To avoid confusion, we have relabeled “offset” as “boundary” in Figure S2.

I'm still confused about offsets generally. Does this maybe mean that the EEG, and each predictor, are all aligned by aligning their endpoints, which are usually/always the ends of sentences? So e.g. all the phoneme activity in the phoneme regressor actually corresponds to those phonemes of the stimuli in the EEG time, but those regressors and EEG do not have a common starting time (one trial to the next maybe?), so they have to be aligned with their ends instead?

We chose to use sentence offsets rather than onsets based on the structure of our input to the HM-LSTM model, where each input consists of a pair of sentences encoded in phonemes, such as “t a_1 n əŋ_2 f ei_1

We understand that it is a bit confusing why the regressor of each level is not aligned to their own offsets in the data. The hidden-layer activations of the HM-LSTM model corresponding to the five linguistic levels (phoneme, syllable, word, phrase, sentence) are consistently 150-dimensional vectors after PCA reduction. As a result, for each input sentence pair, the model produces five distinct hidden-layer activations, each capturing the representational content associated with one linguistic level for the whole sentence. We believe our -100 to 300 ms time window relative to sentence offset reflects a meaningful period during which the brain integrates and comprehends information across multiple linguistic levels.

Being "time-locked to the offset of each sentence at nine latencies" is not something I can really find in any of the references that you mentioned, regarding the offset aspect of this method. Can you point me more specifically to what you are trying to reference with that, or further explain? You said that "predicting EEG signals around the offset of each sentence" is "a method commonly employed in the literature", but the example you gave of Goldstein 2022 is using onsets of words, which is indeed much more in line with what I would expect (not offsets of sentences).

You are correct that Goldstein (2022) aligned model predictions to onsets rather than offsets; however, many studies in the literature also align model predictions with unit offsets. typically because they mark the point at which participants has already processed the relevant information (Brennan, 2016; Brennan et al., 2016; Gwilliams et al., 2024, 2025). Similarly, in our study, we aim to identify neural correlates for each model-derived feature. If we correlate model activity with EEG data aligned to sentence onsets, we would be examining linguistic representations at all levels (from phoneme to sentence) of the whole sentence at the time when participants have not heard the sentence yet. By contrast, aligning to sentence offsets ensures that participants have constructed a full-sentence representation. Although this limits our analysis to a subset of the data (143 sentences × 400 ms windows × 4 conditions), it targets the exact moment when full-sentence representations emerge against background speech, allowing us to examine each model-derived feature onto its neural signature. We have added this clarification on p.12 of the revised manuscript.

This new sentence does not make sense to me: "The regressors are aligned to sentence offsets because all our regressors are taken from the hidden layer of our HM-LSTM model, which generates vector representations corresponding to the five linguistic levels of the entire sentence".

Thank you for the suggestion. We hope our responses in RE4, 15 and 16, along with our supplementary video have now clarified the issue. We have deleted the sentence and provided a more detailed explanation on p.12 of the revised manuscript: The regressors are aligned to sentence offsets because our goal is to identify neural correlates for each model-derived feature of a whole sentence. If we align model activity with EEG data time-locked to sentence onsets, we would be finding neural responses to linguistic levels (from phoneme to sentence) of the whole sentence at the time when participants have not processed the sentence yet. By contrast, aligning to sentence offsets ensures that participants have constructed a full-sentence representation. Although this limits our analysis to a subset of the data (143 sentences × 2 sections × 400 ms windows), it targets the exact moment when full-sentence representations emerge against background speech, allowing us to examine each model-derived feature onto its neural signature. We understand that phonemes, syllables, words, phrases, and sentences differ in their durations. However, the five hidden activity vectors extracted from the model are designed to capture the representations of these five linguistic levels across the entire sentence Specifically, for a sentence pair such as “It can fly

More on the issue of sentence offsets: In response to reviewer 3's question about -100 - 300 ms around sentence offset, you said "Using the entire sentence duration was not feasible, as the sentences in the stimuli vary in length, making statistical analysis challenging. Additionally, since the stimuli consist of continuous speech, extending the time window would risk including linguistic units from subsequent sentence." This does not make sense to me, so can you elaborate? It sounds like you are actually saying that you only analyzed 400 ms of each trial, but that cannot be what you mean.

Yes, we analyzed only the 400 ms window surrounding each sentence offset. Although this represents just a subset of our data (143 sentences × 400 ms × 4 conditions), it precisely captures when full-sentence representations emerge against background speech. Because our model produces a single, condensed representation for each linguistic level over the entire sentence—rather than incrementally—we think it is more appropriate to align to the period surrounding sentence offsets. Additionally, extending the window (e.g. to 2 seconds) would risk overlapping adjacent sentences, since sentence lengths vary. Our focus is on the exact period when integrated, level-specific information for each sentence has formed in the brain, and our results already demonstrate different response patterns to different linguistic levels for the two listener groups within this interval. We have added this clarification on p.13 of the revised manuscript.

In your mTRF analysis, you are now saying that the discrete predictors have "a value of 1" at each of the "boundary offsets", and those TRFs look very similar to your original plots. It sounds to me like you should not be referring to time zero in your original ridge analysis as "sentence offset". If what you mean is that sentence offset time is merely how you aligned the regressors and EEG in time, then your time zero still has a standard, typical TRF interpretation. It is just the point in time, or lag, at which the regressor(s) and EEG are aligned. So activity before zero is "predictive" and activity after zero is "reactive", to think of it crudely. So also in the text, when you say things like "50-150 ms after the sentence offsets", I think this is not really what you mean. I think you are referring to the lags of 50 - 150 ms, relative to the alignment of the regressor and the EEG.

Thank you very much for the explanation. We agree that, in our ridge‐regression time course, pre zero lags index “predictive” processing and post-zero lags index “reactive” processing. Unlike TRF analysis, we applied ridge regression to our high-dimensional model features at nine discrete lags around the sentence offset. At each lag, we tested whether the regression score exceeded a baseline defined as the mean regression score across all lags. For example, finding a significantly higher regression score between 50 and 150 ms suggests that our regressor reliably predicted EEG activity in that time window. So here time zero refers to the precise moment of the sentence offset—not the the alignment of the regressor and the EEG.

I look forward to discussing how much of my interpretation here makes sense or doesn't, both with the authors and reviewers.

Thank you very much for these very constructive feedback and we hope that we have addressed all your questions.